# Integrating Symmetry into Differentiable Planning

## Abstract

We study how group symmetry helps improve data efficiency and generalization for end-to-end differentiable planning algorithms, specifically on 2D robotic path planning problems: navigation and manipulation. We first formalize the idea from Value Iteration Networks (VINs) on using convolutional networks for path planning, because it avoids explicitly constructing equivalence classes and enable end-to-end planning. We then show that value iteration can always be represented as some convolutional form for (2D) path planning, and name the resulting paradigm Symmetric Planner (SymPlan). In implementation, we use steerable convolution networks to incorporate symmetry. Our algorithms on navigation and manipulation, with given or learned maps, improve training efficiency and generalization performance by large margins over non-equivariant counterparts, VIN and GPPN.

## 1   Introduction

Model-based planning usually struggles in complex problems, and planning in more structured and abstract space is a major solution [1, 2, 3, 4]. Symmetry is ubiquitous in learning and decision-making problems and can effectively reduce search space for planning. However, existing planning algorithms using symmetry assumes perfect dynamics knowledge, needs to explicitly build equivalence classes, or does not consider problem structure [5, 4, 6, 7, 8]. For example, if we use A* on path planning, we cannot specify visually obvious rotation symmetry in Figure 1, and need to detect in manually from the provided dynamics model. This would be even more challenging to detect in differentiable planning.

Nevertheless, symmetry in model-free deep reinforcement learning (RL) has been studied recently [9, 10]. However, it can only handle pixel-level "element-wise" symmetry, such as flipping or rotating state and action together. However, a critical benefit of model-free RL agents that enables great asymptotic performance is its end-to-end differentiability. This motivates us to combine the spirit of both: *is it possible to design an end-to-end differentiable planning algorithm that makes use of symmetry in environments?*

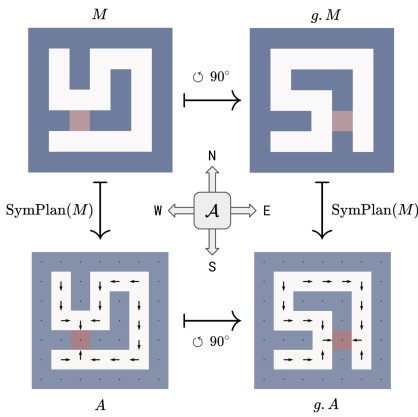

Figure 1: The *path planning problem* has symmetry, so we study how to *exploit* its symmetry in (differentiable) *planning*. Red dots are goal. The optimal actions (arrows) $A = $ `SymPlan`$(M)$ (bottom row) for the maps $M$ (top row) are guaranteed to be equivariant `SymPlan`$(g.M) = g.$`SymPlan`$(M)$ under $\circlearrowleft$ rotations for (2D) path planning. For example, the action in the NW corner of $A$ is the same as the action in the SW corner of $g.A$, after also rotating the arrow $\circlearrowleft 90°$.

In this work, we propose to (1) avoid explicitly building equivalence classes for symmetric states while (2) realize planning in an end-to-end differentiable manner. We are motivated by work in the equivariant network and geometric deep learning community [11, 12, 13, 14, 15, 16], which

treat an RGB image as a mapping $\mathbb{Z}^2 \to \mathbb{R}^3$ and apply equivariant convolutions between feature maps. It satisfies our desiderata: equivariant networks on images do not need to explicitly consider "symmetric pixels" while guarantee symmetry properties. Based on the intuition, we propose a framework, *Symmetric Planning* (SymPlan), to understand a straightforward but general problem, path planning, as operating like images, called steerable feature fields [14, 16]. We focus on 2D grid and prove that value iteration (VI) for 2D path planning is equivariant under the *isometries* of $\mathbb{Z}^2$: translations, rotations, and reflections, and further show that VI here is a special form of steerable convolution network [14]. This provides us a foundation to equip Value Iteration Network (VIN, [17]) with steerable convolution. We implement the equivariant steerable version of VIN, named SymVIN, and use a variant, GPPN, to build SymGPPN. Both SymPlan methods achieve great improvement on training efficiency and generalization performance to unseen random maps, which showcases the advantage of exploiting symmetry from environments for planning.

Our contributions are as follows:

- Understand the inherent symmetry in path planning problems (on 2D grids), formulate value iteration in as steerable convolution network, and connect both to incorporate symmetry into VI.
- Based on the formulation, implement equivariant steerable version of VIN and GPPN.
- Show significant improvement in training and generalization on 2D navigation and manipulation.

## 2    Related work

**Planning with symmetries (Symmetric Planning).** Symmetries widely exist in various domains, and have been exploited in classic planning algorithms as well as model checking [5, 4, 6, 18, 19, 20, 21, 22, 23, 24, 25, 26, 27, 28, 29]. Zinkevich and Balch [7] show the invariance of value function for an MDP with symmetry. Narayanamurthy and Ravindran [8] prove that finding exact symmetry in MDPs is graph isomorphism complete. However, they are based on classic planning algorithms, such as A*, and have a fundamental issue with exploitation of symmetries: they explicitly construct equivalence classes of symmetric states, which explicitly represents states and introduces symmetry breaking. Therefore, they are intractable (NP-hard) in maintaining symmetries in trajectory rollout and forward search (for large state space and symmetry group) and incompatible with differentiable pipelines for representation learning, hindering it from wider applications in RL and robotics.

**State abstraction for detecting symmetries.** Coarsest state abstraction aggregates all symmetric states into equivalence classes, studied in MDP homomorphisms and bisimulation [3, 30, 2]. However, they usually require *perfect* MDP dynamics knowledge and do not scale up well, because of the complexity in maintaining abstraction mappings (homomorphisms) and abstracted MDPs. van der Pol et al. [31] integrate symmetry into model-free RL based on MDP homomorphisms [3], which avoids the challenges in handling symmetry in forward search. Park et al. [32] learn equivariant transition models, but do not consider planning. Additionally, the formulation in commonly defined symmetric MDPs [3, 9, 6, 7] is different from our symmetry formulation for path planning, since they study "element-wise" symmetry for every state-action pairs and require reward to be symmetric. Our reward is not symmetric and we mainly study symmetry of the underlying domain (2D grid), as further discussed in Section B.1.

**Symmetries and equivariance in deep learning.** Equivariant neural networks are used to incorporate symmetry in supervised learning for different domains (e.g. grid and sphere), symmetry groups (e.g. translations and rotations), and group representation on feature spaces [33]. Cohen and Welling [15] introduce G-CNNs, followed by Steerable CNNs [14] which generalizes from scalar feature fields to vector fields with induced representations. Kondor and Trivedi [13], Cohen et al. [12] study theory on the relation between equivariant maps and convolutions. Weiler and Cesa [16] propose to solve kernel constraints under arbitrary representations for $E(2)$ and its subgroups by decomposing into irreducible representations, named $E(2)$-CNN.

**Differentiable planning.** Our pipeline is based on learning to plan in a neural network in a differentiable manner. Value iteration network (VIN) [34] is a representative work that performs value iteration using convolution on lattice grids, and has been further extended [35, 36, 37, 38]. Other than using convolution network, works on integrating learning and planning into differentiable networks include [39, 40, 41, 42, 43, 44, 45, 46, 47, 48, 49]. In the theoretical side, Grimm et al. [50, 51] propose to understand the differentiable planning algorithms from value equivalence perspective, while Gehring et al. [52] study its gradient dynamics.

## 3 Preliminaries

**Markov decision processes.** We model the path planning problems as Markov decision processes (MDP) [1]. An MDP is a 5-tuple $\mathcal{M} = \langle \mathcal{S}, \mathcal{A}, P, R, \gamma \rangle$, with state space $\mathcal{S}$, action space $\mathcal{A}$, transition probability function $P : \mathcal{S} \times \mathcal{A} \times \mathcal{S} \to \mathbb{R}_+$, reward function $R : \mathcal{S} \times \mathcal{A} \to \mathbb{R}$, and discount factor $\gamma \in [0, 1]$. Value functions $V : \mathcal{S} \to \mathbb{R}$ and $Q : \mathcal{S} \times \mathcal{A} \to \mathbb{R}$ represent expected future returns [1].

**Symmetry groups and equivarance.** A symmetry *group* is defined as a set $G$ together with a binary composition map satisfying the axioms of associativity, identity, and inverse. A (left) *group action* of $G$ on a set $\mathcal{X}$ is defined as the mapping $(g, x) \mapsto g.x$ which is compatible with composition. Given a function $f : \mathcal{X} \to \mathcal{Y}$ and $G$ acting on $\mathcal{X}$ and $\mathcal{Y}$, then $f$ is *G-equivariant* if it commutes with group actions: $g.f(x) = f(g.x), \forall g \in G, \forall x \in \mathcal{X}$. In the special case the action on $\mathcal{Y}$ is trivial $g.y = y$, then $f(x) = f(g.x)$ holds, and we say $f$ is *G-invariant*.

**Group representations.** We mainly use two groups: dihedral group $D_4$ and cyclic group $C_4$. The cyclic group of 4 elements is $C_4 = \langle r \mid r^4 = 1 \rangle$, a symmetry group of rotating a square. The dihedral group $D_4 = \langle r, s \mid r^4 = s^2 = (sr)^2 = 1 \rangle$ includes both rotations $r$ and reflections $s$, and has size $|D_4| = 8$. A group representation defines how a group action transforms on a set $G \times S \to S$. These groups have three types of representations of our interest: *trivial*, *regular*, and *quotient* representations, see [16]. The *trivial representation* $\rho_{\text{triv}}$ maps each $g \in G$ to 1 and hence fixes all $s \in S$. The *regular representation* $\rho_{\text{reg}}$ of $C_4$ group sends each $g \in C_4$ to a $4 \times 4$ permutation matrix that cyclically permutes a 4-element vector, such as a one-hot 4-direction action. The regular representation of $D_4$ maps each element to an $8 \times 8$ permutation matrix which does not act on 4-direction actions, which requires the *quotient representations* (quotienting out $fr^2$) and forming a $4 \times 4$ permutation matrix. It is worth mentioning the *standard representation* of the cyclic groups, which are $2 \times 2$ rotation matrices, only used for visualization (Figure 2 middle).

**Steerable feature fields and Steerable CNNs.** The concept of *feature fields* is used in (equivariant) CNNs [11, 12, 13, 14, 15, 16]. The pixels of an 2D RGB image $x : \mathbb{Z}^2 \to \mathbb{R}^3$ on a domain $\Omega = \mathbb{Z}^2$ is a feature field. In steerable CNNs for 2D grid, features are formed as *steerable feature fields* $f : \mathbb{Z}^2 \to \mathbb{R}^C$ that associate a $C$-dimensional feature vector $f(x) \in \mathbb{R}^C$ to each element on a base space, such as $\mathbb{Z}^2$. Defined like this, we know how to transform a steerable feature field and also the feature field after applying CNN on it, using some group [14]. The type of CNNs that operates on steerable feature fields is called Steerable CNN [14], which is equivariant to groups including *translations* as subgroup $(\mathbb{Z}^2, +)$, extending [15]. It needs to satisfy a *kernel steerability* constraint, where the $\mathbb{R}^2$ and $\mathbb{Z}^2$ cases are considered in [16]. We consider the 2D grid as our domain $\Omega = \mathcal{S} = \mathbb{Z}^2$ and use $G = p4m$ group as the running example. The group $p4m = (\mathbb{Z}^2, +) \rtimes D_4$ (wallpaper group) is semi-direct product of discrete translation group $\mathbb{Z}^2$ and dihedral group $D_4$, see [15, 14]. We visualize the *transformation law* of $p4m$ on a feature field on $\Omega = \mathbb{Z}^2$ in **Figure 2** **(Middle)**, usually referred as *induced representation* [14, 16]. Additional details in Section C.

**Planning as convolution.** The core component behind dynamic programming (DP) based algorithms in planning or reinforcement learning is *Bellman (optimality) equation* [53, 1]: $V(s) = \max_a R(s, a) + \gamma \sum_{s'} P(s'|s, a)V(s')$. Value Iteration (VI) iteratively applies Bellman operator and converges to fixed points [1, 53]. The key component of our interest is $\sum_{s'} P(s'|s, a)V(s')$ that aggregates values $V(s')$ from adjacent states by expectation using transition probabilities, here referred as **expected value operation**. Tamar et al. [17] propose Value Iteration Network (VIN) that uses convolution (networks) for planning, as an instance of differentiable planning, by recursively applying planar convolutions and max-pooling over feature spaces on 2D grid $\mathbb{Z}^2$.

## 4 Symmetric Planning Framework

This section formulates the notion of Symmetric Planning (SymPlan). We expand the understanding of path planning in neural networks by planning as convolution on steerable feature fields (*steerable planning*). We use that to build *steerable value iteration* and show it is equivariant.

### 4.1 Steerable Planning: planning on steerable feature fields

We start the discussion based on Value Iteration Networks (VINs, [17]) and use a running example of planning on the 2D grid $\mathbb{Z}^2$. We aim to understand (1) how VIN-style networks embed planning

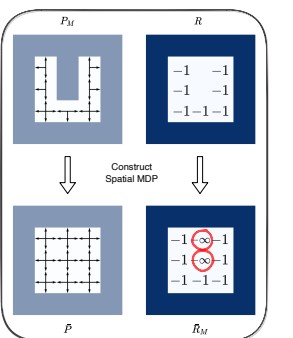 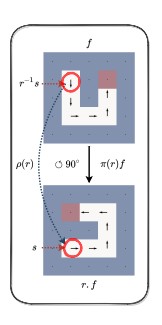 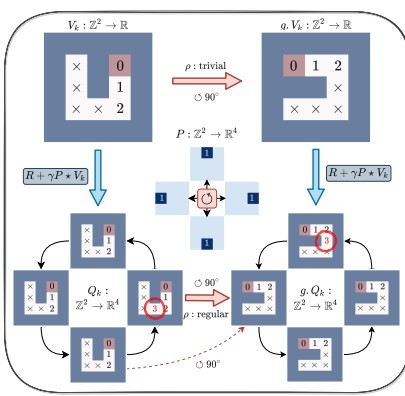

Figure 2: **(Left)** Construction of spatial MDPs from path planning problems, enabling $G$-invariant transition. **(Middle)** The group acts on a feature field (MDP actions). We need to find the element in the original field by $f(r^{-1}x)$, and also rotate the arrow by $\rho(r)$, where $r \in D_4$. We represent one-hot actions as arrows (vector field, using $\rho_{\text{std}}$) for visualization. **(Right)** Equivariance of $V \mapsto Q$ in Bellman operator on feature fields, under $\circlearrowleft 90° \in C_4$ rotation, which visually explains Theorem 4.1. The example simulates VI for one step (see red circles; minus signs omitted) with true transition $P$ using $\circlearrowleft$ N-W-S-E actions. The Q-value field are for 4 actions and can be viewed as either $\mathbb{Z}^2 \to \mathbb{R}^4$ ([14, 16]) or $\mathbb{Z}^2 \rtimes C_4 \to \mathbb{R}$ (on $p4$ group, [15]). See Appendix D for more details.

and how its idea generalizes, (2) how is symmetry structure defined in path planning and how could it be injected into such planning networks.

**Constructing $G$-invariant transition: spatial MDP.** Intuitively, the embedded MDP in a VIN is different from the original path planning problem, since (planar) convolutions are translation equivariant but there are different obstacles in different regions.

We found the key insight in VINs is that it implicitly uses an MDP that has translation equivariance. The core idea behind the construction is that it converts *obstacles* (encoded in transition probability $P$, by *blocking*) into "*traps*" (encoded in reward $\bar{R}$, by $-\infty$ *reward*). This allows to use planar convolutions with translation equivariance, and also enables use to further use steerable convolutions.

The demonstration of the idea is shown in **Figure 2** (Left). We call it *spatial MDP*, with different transition and reward function $\bar{\mathcal{M}} = \langle \mathcal{S}, \mathcal{A}, \bar{P}, \bar{R}_m, \gamma \rangle$, which converts the "complexity" in the transition function $P$ in $\mathcal{M}$ to the reward function $\bar{R}_m$ in $\bar{\mathcal{M}}$. The state and action space are kept the same: state $\mathcal{S} = \mathbb{Z}^2$ and action $\mathcal{A} \subset \mathbb{Z}^2$ to move $\Delta s$ in four directions in a 2D grid. We provide the detailed construction of the spatial MDP in Appendix D.1.

**Steerable features fields.** We generalize the idea from VIN, by viewing functions (in RL and planning) as *steerable feature fields*, motivated by [11, 12, 14]. This is analogous to pixels on images $\Omega \to [255]^3$, and would allow us to apply convolution on it. The state value function is expressed as a field $V : \mathcal{S} \to \mathbb{R}$, while the Q-value function needs a field with $|\mathcal{A}|$ channels: $Q : \mathcal{S} \to \mathbb{R}^{|\mathcal{A}|}$. Similarly, a policy field[1] has probability logits of selecting $|\mathcal{A}|$ actions. For the transition probability $P(s'|s,a)$, we can use action to index it as $P^a(s'|s)$, similarly for reward $R^a(s)$. The next section will show that we can convert the transition function to field and even convolutional filter. Additional details are in Appendix D.

### 4.2 Symmetric Planning: symmetry by equivariance

The seemingly slight change in the construction of spatial MDPs brings important symmetry structure. The general idea in exploiting symmetry in path planning is to use *equivariance* to avoid explicitly constructing equivalence classes of symmetric states. To this end, we construct value iteration over steerable feature fields, and show it is *equivariant* for path planning.

In VIN, the convolution is over 2D grid $\mathbb{Z}^2$, which is symmetric under $D_4$ (rotations and reflections). However, we also know that VIN is already equivariant under translations. To consider all symmetries, as in [14, 16], we understand the group $p4m = G = B \rtimes H$ as constructed by a *base space* $B = G/H = (\mathbb{Z}^2, +)$ and a *fiber* group $H = D_4$, which is a *stabilizer subgroup* that fixes the origin $\mathbf{0} \in \mathbb{Z}^2$. We could then formally study such symmetry in the spatial MDP, since we construct it to

---

[1]We avoid the symbol $\pi$ for policy since it is used for induced representation in [14, 16].

ensure that the transition probability function in $\bar{\mathcal{M}}$ is $G$-invariant. Specifically, we can uniquely decompose any $g \in \mathbb{Z}^2 \rtimes D_4$ as $t \in \mathbb{Z}^2$ and $r \in D_4$ (and translations act "trivially" on action), so

$$\bar{P}(s' \mid s, a) = \bar{P}(g.s' \mid g.s, g.a) \equiv \bar{P}\left((tr).s' \mid (tr).s, r.a\right), \quad \forall g = tr \in \mathbb{Z}^2 \rtimes D_4, \forall s, a, s'. \quad (1)$$

**Expected value operator as steerable convolution.** The equivariance property can be shown step-by-step: (1) *expected value operation*, (2) *Bellman operator*, and (3) full *value iteration*. First, we use $G$-invariance to prove that the expected value operator $\sum_{s'} P(s'|s,a)V(s')$ is equivariant.

**Theorem 4.1.** *If transition is $G$-invariant, the expected value operator $E$ over $\mathbb{Z}^2$ is $G$-equivariant.*

The proof is in Appendix E.1 and visual understanding is in Figure 2 left. However, this provides intuition but is inadequate since we do not know: (1) how to implement it with CNNs, (2) how to use multiple feature channels like VINs, since it shows for scalar-valued transition probability and value function (corresponding to trivial representation). To this end, we next prove that we can implement value iteration using steerable convolution with general steerable kernels.

**Theorem 4.2.** *If transition is $G$-invariant, there exists a (one-argument, isotropic) matrix-valued steerable kernel $P^a(s - s')$ (for every action), such that the expected value operator can be written as a steerable convolution and is $G$-equivariant:*

$$E[V] = P^a \star V, \quad [g.[P^{g.a} \star V]](s) = [P^a \star [g.V]](s), \quad \forall s \in \mathbb{Z}^2, \forall g \in \mathbb{Z}^2 \rtimes D_4. \quad (2)$$

The full derivation is provided in Appendix E. We write the transition probability as $P^a(s, s')$, and we show it only depends on *state difference* $P^a(s - s')$ (or *one-argument* kernel [12]) using $G$-invariance, which is the key step to show it is some *convolution*. Note that we use one kernel $P^a$ for each action (four directions), and when the group acts on $E$, it also acts on the action $P^{g.a}$ (and state, so technically acting on $\mathcal{S} \times \mathcal{A}$). Additionally, if the steerable kernel also satisfies the $D_4$-*steerability constraint* [16, 54], the steerable convolution is *equivariant* under $p4m = \mathbb{Z}^2 \rtimes D_4$. We can then extend VINs from $\mathbb{Z}^2$ translation equivariance to $p4m$-equivariance (translations, rotations, reflections). The derivation follows the existing work on steerable CNNs [15, 14, 16, 12], while this is our goal: to justify the close connection between path planning and steerable convolutions.

**Steerable Bellman operator and value iteration.** We can now represent all operations in Bellman (optimality) operator on steerable feature fields over $\mathbb{Z}^2$ (or *steerable Bellman operator*) as follows:

$$V_{k+1}(s) = \max_a R^a(s) + \gamma \times [P^a \star V_k](s), \quad (3)$$

where $V, R^a, \bar{P}^a$ are steerable feature fields over $\mathbb{Z}^2$. As for the operations, $\max_a$ is (max) pooling (over group channel), $+, \times$ are point-wise operations, and $\star$ is convolution. As the second step, the main idea is to prove every operation in Bellman (optimality) operator on steerable fields is equivariant, including the nonlinear $\max_a$ operator and $+, \times$. Then, iteratively applying Bellman operator forms value iteration and is also equivariant, as shown below and proved in Appendix E.4.

**Proposition 4.3.** *For a spatial MDP with $G$-invariant transition, the optimal value function can be found through $G$-steerable value iteration.*

**Remark.** Our framework generalizes the idea behind VINs and enables us to understand its applicability and restrictions. More importantly, this allows us to integrate symmetry but avoid explicitly building equivalence classes and enables planning with symmetry in end-to-end fashion. We emphasize that the *symmetry in spatial MDPs* is different from *symmetric MDPs* [7, 3, 9], since our reward function is *not* $G$-invariant. Although we focus on $\mathbb{Z}^2$, we can generalize to path planning on higher-dimensional or even continuous Euclidean spaces (like $\mathbb{R}^3$ space [54] or spatial graphs in $\mathbb{R})^3$ [55]), and use *equivariant operations* on *steerable feature fields* (such as steerable convolutions, pooling, and point-wise non-linearities) from steerable CNNs. We refer the readers to Appendix D and to [15, 14, 56, 16] for more details.

## 5 Symmetric Planning in Practice

In this section, we discuss how to achieve Symmetric Planning on 2D grids with E(2)-steerable CNNs [16]. We focus on implementing symmetric version of value iteration, SymVIN, and generalize the methodology to make a symmetric version of a popular follow-up of VIN, GPPN [36].

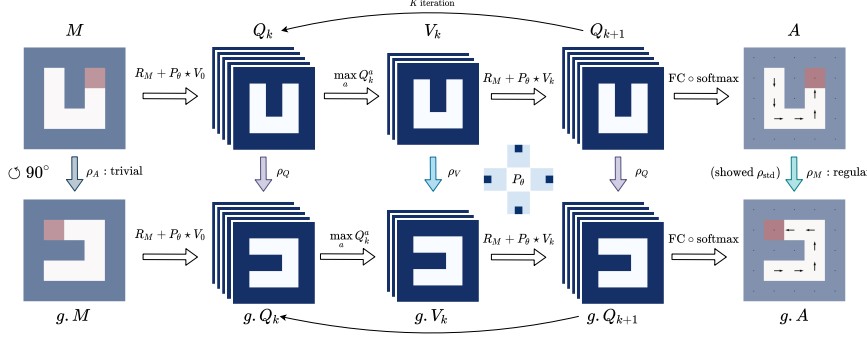

Figure 3: Commutative diagram for the full pipeline of SymVIN on steerable feature fields over $\mathbb{Z}^2$ (every grid). If rotating the input map $M$ by $\pi_M(g)$ of any $g$, the output action $A = \texttt{SymVIN}(M)$ is guaranteed to be transformed by $\pi_A(g)$, i.e. the entire steerable SymVIN is equivariant under induced representations $\pi_M$ and $\pi_A$: $\texttt{SymVIN}(\pi_M(g)M) = \pi_A(g)\texttt{SymVIN}(M)$. We use stacked feature fields to emphasize that SymVIN supports direct-sum of representations beyond scalar-valued.

**Steerable value iteration.** We have showed that, value iteration for path planning problems on $\mathbb{Z}^2$ consists of equivariant maps between steerable feature fields. It can be implemented as an equivariant steerable CNN, with recursively applying two alternating (equivariant) layers:

$$Q_k^a(s) = R_m^a(s) + \gamma \times [P_\theta^a \star V_k](s), \quad V_{k+1}(s) = \max_a Q_k^a(s), \quad s \in \mathbb{Z}^2, \tag{4}$$

where $k \in [K]$ indexes iteration, $V_k, Q_k^a, R_m^a$ are steerable feature fields over $\mathbb{Z}^2$ output by equivariant layers, $P_\theta^a$ is a learned kernel in neural network, and $+, \times$ are element-wise operations.

**Pipeline.** We follow the pipeline in VIN [17]. The commutative diagram for the full pipeline is shown in Figure 3. The path planning task is given by a $m \times m$ spatial binary obstacle occupancy map and one-hot goal map, represented as a feature field $M : \mathbb{Z}^2 \rightarrow \{0,1\}^2$. For the iterative process $Q_k^a \mapsto V_k \mapsto Q_{k+1}^a$, the reward field $R_M$ is predicted from map $M$ (by a $1 \times 1$ convolution layer) and the value field $V_0$ is initialized as zeros. The network output is (logits of) planned actions for all locations[2], represented as $A : \mathbb{Z}^2 \rightarrow \mathbb{R}^{|\mathcal{A}|}$, predicted from the final Q-value field $Q_K$ (by another $1 \times 1$ convolution layer). The number of iterations $K$ and the convolutional kernel size $F$ of $P_\theta^a$ are set based on map size $M$, and the spatial dimension $m \times m$ is kept consistent.

**Building Symmetric Value Iteration Networks.** Given the pipeline of VIN fully on steerable feature fields, we are ready to build equivariant version with E(2)-steerable CNNs [16]. The idea is to replace every Conv2d with a steerable convolution layer between steerable feature fields, and associate the fields with proper fiber representations $\rho(h)$.

VINs use ordinary CNNs and can choose the size of intermediate feature maps. The design choices in steerable CNNs is the feature fields and fiber representations (or *type*) for every layer [14, 16]. The main difference[3] in steerable CNNs is that we also need to tell the network how to *transform* every *feature field*, by specifying *fiber representations*, as shown in Figure 3.

**Specification of input map and output action.** We first specify *fiber representations* for the input and output field of the network: map $M$ and action $A$. For input **occupancy map and goal** $M : \mathbb{Z}^2 \rightarrow \{0,1\}^2$, it does not $D_4$ to act on the 2 channels, so we use two copies of trivial representations $\rho_M = \rho_{\text{triv}} \oplus \rho_{\text{triv}}$. For **action**, the final action output $A : \mathbb{Z}^2 \rightarrow \mathbb{R}^{|\mathcal{A}|}$ is for logits of four actions $\mathcal{A} = (\texttt{north}, \texttt{west}, \texttt{south}, \texttt{east})$ for every location. If we use $H = C_4$, it naturally acts on the four actions (ordered $\circlearrowleft$) by *cyclically* $\circlearrowleft$ *permuting* the $\mathbb{R}^4$ channels. However, since the $D_4$ group has 8 elements, we need a *quotient representation*, see [16] and Appendix F.

**Specification of intermediate fields: value and reward.** Then, for the intermediate feature fields: Q-values $Q_k$, state value $V_k$, and reward $R_m$, we are free to choose fiber representations, as well as the width (number of copies). For example, if we want 2 copies of regular representation of $D_4$, the feature field has $2 \times 8 = 16$ channels and the stacked representation is $16 \times 16$ (by direct-sum).

For the **Q-value field** $Q_k^a(s)$, we use representation $\rho_Q$ and its size as $C_Q$. We need at least $C_A \geq |\mathcal{A}|$ channels for all actions of $Q(s,a)$ as in VIN and GPPN, then stacked together and denoted as $Q_k \triangleq \bigoplus_a Q_k^a$ with dimension $Q_k : \mathbb{Z}^2 \rightarrow \mathbb{R}^{C_Q * C_A}$. Therefore, the representation is direct-sum

---

[2]Technically, it also includes values or actions for obstacles, since the network needs to learn to approximate the reward $R_M(s, \Delta s) = -\infty$ with enough small reward and avoid obstacles.

$\bigoplus \rho_Q$ for $C_A$ copies. The **reward** is implemented similarly as $R_M \triangleq \bigoplus_a R_M^a$ and must have same dimension and representation to add element-wisely. For **state value** field, we denote the choose as fiber representation as $\rho_V$ and its size $C_V$. It has size $V_k : \mathbb{Z}^2 \to \mathbb{R}^{C_V}$ Thus, the steerable kernel is *matrix-valued* with dimension $P_\theta : \mathbb{Z}^2 \to \mathbb{R}^{(C_Q * C_A) \times C_V}$. In practice, we found using *regular representations* for all three works the best. It can be viewed as "augmented" state and is related to group convolution, detailed in Appendix F.

**Other operations.** We now visit the remained (equivariant) operations. (1) The $\max$ **operation in** $Q_k \mapsto V_{k+1}$. While we have showed the $\max$ operation in $V_{k+1}(s) = \max_a Q_k^a(s)$ is equivariant in Theorem 4.3, we need to apply $\max$(-pooling) for all actions along the "representation channel" from stacked representations $C_A * C_Q$ to one $C_Q$. More details are in Appendix F.5. (2) The **final output layer** $Q_K \mapsto A$. After the final iteration, the $Q$-value field $Q_k$ is fed into the policy layer with $1 \times 1$ convolution to convert the action logit field $\mathbb{Z}^2 \to \mathbb{R}^{|\mathcal{A}|}$.

**Extended method: Symmetric GPPN.** Gated path planning network (GPPN [36]) proposes to use LSTM to alleviate the issue of unstable gradient in VINs. Although it does not strictly follow value iteration, it still follows the spirit of steerable planning. Thus, we first obtained a fully convolutional variant of GPPN from [Redacted for anonymous review], called ConvGPPN. It replaces the MLPs in the original LSTM cell with convolutional layers, and then replaces convolutions with equivariant steerable convolutions, resulting in a fully equivariant SymGPPN. See Appendix F.3 for details.

**Extended tasks: planning on learned maps with mapper networks.** We consider two planning tasks on 2D grids: 2D navigation and 2-DOF manipulation. To demonstrate the ability of handling symmetry in differentiable planning, we consider more complicated state space input: visual navigation and workspace manipulation, and discuss how to use mapper networks to convert the state input and use end-to-end learned maps, as in [36, 37]. See Appendix F.2 for details.

# 6 Experiments

We experiment VIN, GPPN and our SymPlan methods on four path planning tasks, including using *given* or *learned* maps. The additional experiments and ablation studies are in Appendix G.

**Environments and datasets.** We demonstrate the idea in two major robotics tasks: *navigation* and *manipulation*. We focus on the 2D regular grid setting for path planning, as adopted in prior work [17, 36, 37]. For each task, we consider using either *given* (2D navigation and 2-DOF configuration-space manipulation) or *learned* maps (visual navigation and 2-DOF workspace manipulation). In the latter case, the planner needs to jointly learn a mapper that converts egocentric panoramic images (visual navigation) or workspace states (workspace manipulation) into plannable loss, as in [36, 37]. In both cases, we randomly generate training, validation and test data of $10K/2K/2K$ maps for all map sizes, to demonstrate data efficiency and generalization ability of symmetric planning. Note that the test maps are unlikely to be symmetric to the training maps by any transformation from the symmetry groups $G$. For all environments, the planning domain is the 2D regular grid $\mathcal{S} = \Omega = \mathbb{Z}^2$, and the action space is to move in 4 $\circlearrowleft$ directions[4]: $\mathcal{A} = (\texttt{north}, \texttt{west}, \texttt{south}, \texttt{east})$.

**Methods: planner networks.** We compare five planner methods, where two are our SymPlan version of their non-equivariant counterparts. Our equivariant implementation is based on *Value Iteration Networks* (**VIN**, [17]) and *Gated Path Planning Networks* (**GPPN**, [36]). We implement the equivariant version of VIN, named **SymVIN**. For GPPN, we first obtained a *fully convolutional* version, named **ConvGPPN** [Redacted for anonymous review], and furthermore **SymGPPN** with steerable CNNs. All methods use (equivariant) convolutions with *circular padding* in planning in configuration spaces for the manipulation tasks, except GPPN that is not fully convolutional. Chaplot et al. [37] propose SPT based on Transformers, while integrating symmetry to Transformers is beyond steerable convolutions, thus we do not consider it but still adopt some useful setup.

**Training and evaluation.** We report successful rate and its training curves over 3 seeds for each setup. The training process of the given map setup follows [17, 36], where we train 30 epochs with batch size 32, and use kernel size $F = 3$ by default. The gradient clip threshold is set to 5. The

---

[4]Note that the MDP action space $\mathcal{A}$ needs to be *compatible* with the group action $G \times \mathcal{A} \to \mathcal{A}$. Since the E2CNN package [16] uses *counterclockwise* rotations $\circlearrowleft$ as generators for rotation groups $C_n$, the action space needs to be *counterclockwise* $\circlearrowleft$.

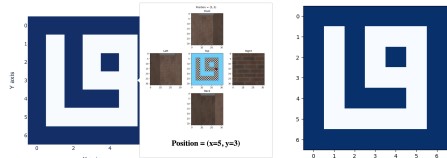 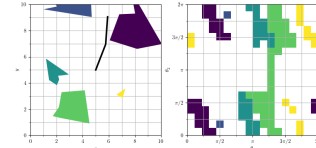

Figure 4: **(Left)** A visual navigation environment rendered from a randomly generated $7 \times 7$ maze **(Middle)**, where the hover is the visualization of four views at position $(5, 3)$. **(Right)** A 2-joint manipulation task in workspace (topdown) and configuration space (2 DOFs) in $18 \times 18$ resolution.

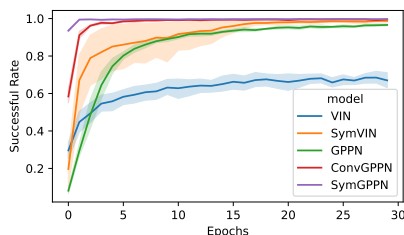 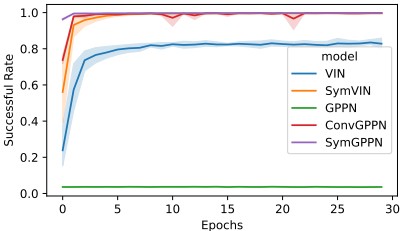

Figure 5: Training curves on **(Left)** 2D navigation with 10K of $15 \times 15$ maps and on **(Right)** 2DoFs manipulation with 10K of $18 \times 18$ maps in configuration space. Faded areas indicate standard error.

default batch size is 32, while we need to reduce for some GPPN variants, since LSTM consumes much more memory.

## 6.1  Planning on given maps

**Environmental setup.** In the **2D navigation** task, the map and goal are randomly generated, where the map size is $\{15, 28, 50\}$. In **2-DOF manipulation** in configuration space, we adopt the setting in [37] and train networks to take as input of configuration space, represented by two joints. We randomly generate 0 to 5 obstacles in the manipulator workspace. Then the 2 degree-of-freedom (DOF) configuration space is constructed from workspace and discretized into 2D grid with sizes $\{18, 36\}$, corresponding to bins of $20°$ and $10°$, respectively. All methods are trained using the same network size, where for equivariant versions, we use *regular* representations for all layers, which has size $|D_4| = 8$. We keep the same parameters for all methods, so all equivariant convolution layers with *regular* representations will have higher embedding sizes. Due to memory constraint, we use $K = 30$ iterations for 2D maze navigation, and $K = 27$ for manipulation. We use kernel sizes $F = \{3, 5, 5\}$ for $m = \{15, 28, 50\}$ navigation, and $F = \{3, 5\}$ for $m = \{18, 36\}$ manipulation.

**Results.** We show the averaged test results for both 2D navigation and C-space manipulation tasks on generalizing to unseen maps (Table 1) and the training curves for all methods (Figure 5). For VIN series, our SymVIN is much better than the vanilla VIN in terms of generalization and training performance in both environments, which learns much faster and achieves almost perfect asymptotic performance. As for GPPN, we found the fully convolutional variant ConvGPPN actually works better than the original one in [36], especially in learning speed. SymGPPN further boosts ConvGPPN and outperforms all other methods, including our SymVIN. One exception is GPPN learns poorly in C-space manipulation. For GPPN, the added circular padding in the convolution encoder leads to gradient vanishing problem.

Additionally, we found using regular representations (for $D_4$ or $C_4$) for state value $V : \mathbb{Z}^2 \to \mathbb{R}^{C_V}$ (and for $Q$-value) works better than trivial representations. This is counterintuitive since we expect the $V$ value to be scalar $\mathbb{Z}^2 \to \mathbb{R}$. One reason is that switching between regular (for $Q$) and trivial (for $V$) representation introduces unnecessary bottleneck. Depending on the choose of representations, we implement different max-pooling, with details in Appendix F.5. We also empirically found using FC only in the final layer $Q_K \mapsto A$ helps stabilize the training a bit. The ablation study on this and more are in Appendix G.

**Remark.** Two symmetric planners are both significantly better than their counterparts. Notably, we did not include any symmetric maps to the test data that symmetric planners would perform much better. There are several potential sources of advantages: (1) SymPlan allows parameter sharing across positions and maps and implicitly enables planning in a reduced space: every $(s, a, s')$

Table 1: Averaged test success rate (%) for using 10K/2K/2K dataset for all four types of tasks.

| Method | Navigation | | | | Manipulation | | |
|---|---|---|---|---|---|---|---|
| (10K Data) | $15 \times 15$ | $28 \times 28$ | $50 \times 50$ | Visual | $18 \times 18$ | $36 \times 36$ | Workspace |
| VIN | 66.97 | 67.57 | 57.92 | 50.83 | 77.82 | 84.32 | 80.44 |
| **SymVIN** | 98.99 | 98.14 | 86.20 | 95.50 | 99.98 | 99.36 | **91.10** |
| GPPN | 96.36 | 95.77 | 91.84 | 93.13 | 2.62 | 1.68 | 3.67 |
| ConvGPPN | 99.75 | 99.09 | 97.21 | 98.55 | 99.98 | 99.95 | 89.88 |
| **SymGPPN** | **99.98** | **99.86** | **99.49** | **99.78** | **100.00** | **99.99** | 90.50 |

seamlessly generalizes to $(g.s, g.a, g.s')$ for any $g \in G$, (2) thus it uses training data more efficiently, (3) it reduces the space of hypothesis class and facilitate generalization to unseen maps.

## 6.2 Planning on learned maps: simultaneously planning and mapping

**Environmental setup.** For **visual navigation**, we randomly generate maps using the same strategy as before, and then render four egocentric panoramic views for each location from produced 3D environments with *Gym-MiniWorld* [57], since it allows to generate 3D mazes with any layout. For $m \times m$ maps, all egocentric views for a map is represented by $m \times m \times 4$ RGB images. For **workspace manipulation**, we randomly generate 0 to 5 obstacles in workspace as before. We use a mapper network to convert the $96 \times 96$ workspace (image of obstacles) to the $m \times m$ 2 degree-of-freedom (DOF) configuration space (2D occupancy grid). In both environments, the setup is similar to Section 6.1, while we only use $m = 15$ maps but longer 100 epochs for visual navigation and $m = 18$ maps still with 30 epochs for workspace manipulation.

**Methods: mapper networks and setup.** For **visual navigation**, we follow the mapper network setup in [36]. A mapper network converts every image into a 256-dimensional embedding $m \times m \times 4 \times 256$ and then predicts map layout $m \times m \times 1$. For **workspace manipulation**, we use U-net [58] with residual-connection [59] as a mapper. See Section G for details.

**Results.** The results are also shown in Table 1, denoted as Visual (navigation, $15 \times 15$) and Workspace (manipulation, $18 \times 18$). In visual navigation, the trends are similar to 2D case: two symmetric planners both train much faster. Besides vanilla VIN, all approaches finally converge to near-optimal successful rate (around $95\%$), while the validation and test results show large gaps. SymGPPN has almost no generalization gap, while VIN does not generalize well to new 3D visual navigation environments. Our SymVIN improves test successful rate from less than $50\%$ to $90\%$ and is comparable with GPPN. Since the input is raw images and a mapper is used to learn end-to-end, it potentially causes one major source of generalization gap for some approaches. In workspace manipulation, the results are also analogous to C-space, while ours advantages over baselines are smaller. In our inspection, we found the mapper network is the bottleneck, since the mapping for obstacles from workspace to C-space is nontrivial to learn.

**Remark.** The SymPlan models demonstrate end-to-end planning and learning ability, potentially enabling further applications to other tasks as a differentiable component for planning. The additional results and ablation studies are provided in Appendix G.

## 7 Discussion

In this work, we study the symmetry in 2D path planning problem, and build a framework using the theory of steerable CNNs to prove that value iteration in path planning is actually a form of steerable CNN (on 2D grids). Although we focus on $\mathbb{Z}^2$, we can generalize to path planning on higher-dimensional or even continuous Euclidean spaces [54, 55], and use *equivariant operations* on *steerable feature fields* (such as steerable convolutions, pooling, and point-wise non-linearities) from steerable CNNs. We practically show that the SymPlan algorithms exactly motivated by the theory provide great improvement. We hope the framework along with the design of practical algorithm can enable new perspective to exploit the symmetry structure in path planning problems. Although it still has some limitations, such as (1) that the action needs to be known as moving on the domain (2D), and (2) that it is not sampling-based and may struggle for high-dimensional problems, we believe that it has potential extensions to more general formulation.

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
