**Training and evaluation.** We report success rate and training curves over 3 seeds. The training process (on given maps) follows [17, 18], where we train 30 epochs with batch size 32, and use kernel size $F = 3$ by default. The gradient clip threshold is set to 5. The default batch size is 32, while we need to reduce for some GPPN variants, since LSTM consumes much more memory.

## 6.1 Planning on given maps

**Environmental setup.** In the **2D navigation** task, the map and goal are randomly generated, where the map size is $\{15, 28, 50\}$. In **2-DOF manipulation** in configuration space, we adopt the setting in [37] and train networks to take as input of configuration space, represented by two joints. We randomly generate 0 to 5 obstacles in the manipulator workspace. Then the 2 degree-of-freedom (DOF) configuration space is constructed from workspace and discretized into 2D grid with sizes $\{18, 36\}$, corresponding to bins of $20°$ and $10°$, respectively. All methods are trained using the same network size, where for equivariant versions, we use *regular* representations for all layers, which has size $|D_4| = 8$. We keep the same parameters for all methods, so all equivariant convolution layers with *regular* representations will have higher embedding sizes. Due to memory constraint, we use $K = 30$ iterations for 2D maze navigation, and $K = 27$ for manipulation. We use kernel sizes $F = \{3, 5, 5\}$ for $m = \{15, 28, 50\}$ navigation, and $F = \{3, 5\}$ for $m = \{18, 36\}$ manipulation.

**Results.** We show the averaged test results for both 2D navigation and C-space manipulation tasks on generalizing to unseen maps (Table 1) and the training curves for all methods (Figure 5). For VIN series, our SymVIN is much better than the vanilla VIN in terms of generalization and training performance in both environments, which learns much faster and achieves almost perfect asymptotic performance. As for GPPN, we found the fully convolutional variant ConvGPPN actually works better than the original one in [18], especially in learning speed. However, SymVIN does fluctuate in some runs, which seems to come from initialization and label, further studied in Appendix. SymGPPN further boosts ConvGPPN and outperforms all other methods. One exception is GPPN learns poorly in C-space manipulation. For GPPN, the added circular padding in the convolution encoder leads to gradient vanishing problem.

Additionally, we found using regular representations (for $D_4$ or $C_4$) for state value $V : \mathbb{Z}^2 \to \mathbb{R}^{C_V}$ (and for $Q$-value) works better than trivial representations. This is counterintuitive since we expect the $V$ value to be scalar $\mathbb{Z}^2 \to \mathbb{R}$. One reason is that switching between regular (for $Q$) and trivial (for $V$) representation introduces unnecessary bottleneck. Depending on the choice of representations, we implement different max-pooling, with details in Appendix J.5. We also empirically found using FC only in the final layer $Q_K \mapsto A$ helps stabilize the training a bit. The ablation study on this and more are in Appendix E.

Table 1: Averaged test success rate (%) for using 10K/2K/2K dataset for all four types of tasks.

| Method (10K Data) | Navigation | | | Visual | Manipulation | | Workspace |
|---|---|---|---|---|---|---|---|
| | $15 \times 15$ | $28 \times 28$ | $50 \times 50$ | | $18 \times 18$ | $36 \times 36$ | |
| VIN | 66.97 | 67.57 | 57.92 | 50.83 | 77.82 | 84.32 | 80.44 |
| **SymVIN** | 98.99 | 98.14 | 86.20 | 95.50 | 99.98 | 99.36 | **91.10** |
| GPPN | 96.36 | 95.77 | 91.84 | 93.13 | 2.62 | 1.68 | 3.67 |
| ConvGPPN | 99.75 | 99.09 | 97.21 | 98.55 | 99.98 | 99.95 | 89.88 |
| **SymGPPN** | **99.98** | **99.86** | **99.49** | **99.78** | **100.00** | **99.99** | 90.50 |

**Remark.** Two symmetric planners are both significantly better than their counterparts. Notably, we did not include any symmetric maps to the test data that symmetric planners would perform much better. There are several potential sources of advantages: (1) SymPlan allows parameter sharing across positions and maps and implicitly enables planning in a reduced space: every $(s, a, s')$ seamlessly generalizes to $(g.s, g.a, g.s')$ for any $g \in G$, (2) thus it uses training data more efficiently, (3) it reduces the space of hypothesis class and facilitate generalization to unseen maps.

### 6.2 Planning on learned maps: simultaneously planning and mapping

**Environmental setup.** For **visual navigation**, we randomly generate maps using the same strategy as before, and then render four egocentric panoramic views for each location from produced 3D environments with *Gym-MiniWorld* [57], since it allows to generate 3D mazes with any layout. For $m \times m$ maps, all egocentric views for a map is represented by $m \times m \times 4$ RGB images. For **workspace manipulation**, we randomly generate 0 to 5 obstacles in workspace as before. We use a mapper network to convert the $96 \times 96$ workspace (image of obstacles) to the $m \times m$ 2 degree-of-freedom (DOF) configuration space (2D occupancy grid). In both environments, the setup is similar to Section 6.1, while we only use $m = 15$ maps but longer 100 epochs for visual navigation and $m = 18$ maps still with 30 epochs for workspace manipulation.

**Methods: mapper networks and setup.** For **visual navigation**, we follow the mapper network setup in [18]. A mapper network converts every image into a 256-dimensional embedding $m \times m \times 4 \times 256$ and then predicts map layout $m \times m \times 1$. For **workspace manipulation**, we use U-net [58] with residual-connection [59] as a mapper. See Section E for details.

**Results.** The results are also shown in Table 1, denoted as Visual (navigation, $15 \times 15$) and Workspace (manipulation, $18 \times 18$). In visual navigation, the trends are similar to 2D case: two symmetric planners both train much faster. Besides vanilla VIN, all approaches finally converge to near-optimal successful rate (around $95\%$), while the validation and test results show large gaps. SymGPPN has almost no generalization gap, while VIN does not generalize well to new 3D visual navigation environments. Our SymVIN improves test successful rate from less than $50\%$ to $90\%$ and is comparable with GPPN. Since the input is raw images and a mapper is used to learn end-to-end, it potentially causes one major source of generalization gap for some approaches. In workspace manipulation, the results are also analogous to C-space, while ours advantages over baselines are smaller. In our inspection, we found the mapper network is the bottleneck, since the mapping for obstacles from workspace to C-space is nontrivial to learn.

**Remark.** The SymPlan models demonstrate end-to-end planning and learning ability, potentially enabling further applications to other tasks as a differentiable component for planning. The additional results and ablation studies are provided in Appendix E.

## 7 Discussion

In this work, we study the symmetry in 2D path planning problem, and build a framework using the theory of steerable CNNs to prove that value iteration in path planning is actually a form of steerable CNN (on 2D grids). Although we focus on $\mathbb{Z}^2$, we can generalize to path planning on higher-dimensional or even continuous Euclidean spaces [54, 55], and use *equivariant operations* on *steerable feature fields* (such as steerable convolutions, pooling, and point-wise non-linearities) from steerable CNNs. We practically show that the SymPlan algorithms exactly motivated by the theory provide great improvement. We hope the framework along with the design of practical algorithms can provide a new pathway to exploiting symmetry structure in differentiable planning.

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

## A  Outline

The appendix is organized as follows. Blue text highlights new content in revision for rebuttal.

1. A temporary section for new figures and results for rebuttal.
2. A preliminary version of a section on Symmetric Planning with less prerequisites on equivariant networks.
3. Additional experimental setup and empirical results. (This section is moved here previously at the end of appendix.)
4. Discussion on the considered symmetry, as well as limitations and extensions.
5. Additional technical background and concepts on steerable CNNs and group CNNs, useful for understanding how to apply our setup to other problems and setup.
6. More details and interpretation on the steerable planning framework.
7. Full derivation and proofs.
8. Other practice implementation details.

## B  A temporary section for new figures and results

This temporary section is a collection of all new figures and results for the rebuttal purpose. All the content will be merged into the corresponding sections in the future version.

### B.1  Updated environment figures

To emphasize the tasks, we update the figures for the environments in our experiments, along with demonstration of the learned model.

We show the figures for **Configuration-space and Workspace manipulation** in Figure 6, and the figures for **2D and Visual Navigation** in Figure 7.

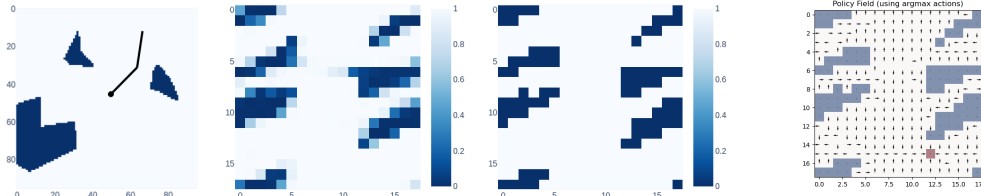

Figure 6: A set of visualization for a 2-joint manipulation task. The obstacles are randomly generated. **(1)** The 2-joint manipulation task shown in top-down workspace with $96 \times 96$ resolution. This is used as the input to the **Workspace Manipulation** task. **(2)** The predicted configuration space in resolution $18 \times 18$ from a mapper module, which is jointly optimized with a planner network. **(3)** The ground truth configuration space from a handcraft algorithm in resolution $18 \times 18$. This is used as input to the **Configuration-space (C-space) Manipulation** task and as target in the auxiliary loss for the Workspace Manipulation task (as done in SPT [37]). **(4)** The predicted policy (overlaid with C-space obstacle for visualization) from an end-to-end trained SymVIN model that uses a mapper to take the top-down workspace image and plans on a learned map. The red block is the goal position.

### B.2  Results on generalization to larger maps

To better demonstrate the empirical difference, we conduct new experiment on generalization to larger maps. We hope this can alleviate some concern on (1) scalability and (2) performance gap between SymGPPN and ConvGPPN.

We experiment all methods on map size $15 \times 15$ through $99 \times 99$, averaging over 3 seeds (3 model checkpoints) for each method and 1000 maps for each size. Note that all models are trained on $15 \times 15$ with $K = 30$.

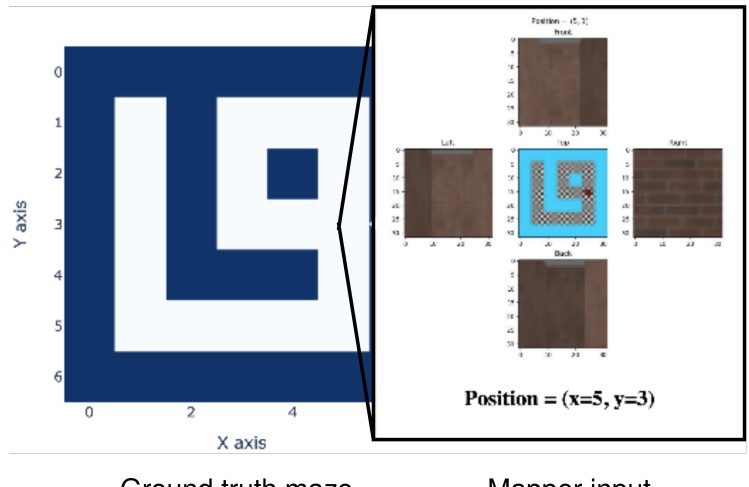

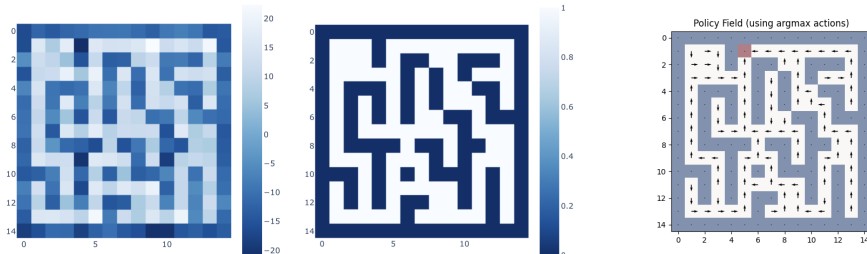

Figure 7: A set of visualization for 2D navigation and visual navigation. The maze is randomly generated. **(1, top)** The 3D visual navigation environment generated by an illustrative $7 \times 7$ map, where we highlight the panoramic view at a position $(5, 3)$ with four RGB images (resolution $32 \times 32 \times 3$). The entire observation tensor for this $7 \times 7$ example visual navigation environment is $7 \times 7 \times 4 \times 32 \times 32 \times 3$. This is used as the input to the **Visual Navigation** task. **(2)** Another predicted map in resolution $15 \times 15$ from a mapper module, which is jointly optimized with a planner network. We show the visualization a different map used in actual training. **(3)** The ground truth map in resolution $15 \times 15$. This is also used as input to the **2D Navigation** task and as target in the auxiliary loss for the Visual Navigation task (as done in GPPN). **(4)** The predicted policy from an end-to-end trained SymVIN model that uses a mapper to take the observation images (formed as a tensor) and plans on a learned map. The red block is the goal position.

Between $15 \times 15$ and $49 \times 49$ we use all odd-size maps, and between $51 \times 51$ and $99 \times 99$ we use interval of 4 ($51 \times 51 \rightarrow 55 \times 55$ ...). We only use odd size maps because for the even size maps the maze generation algorithm would cause non-symmetric pattern (missing right and bottom boundary).

Note that we disable backward pass during evaluation. However, we observe that GPPN variants do use much more memory if backward pass is enabled because (1) they rely on the costly computation of LSTM, and (2) the number of parameters is also significantly larger. The training and inference time used by GPPN variants is also significantly longer. We omit the consideration of resource and time issue and focus on the final generalization results.

We focus on comparing SymPlan methods with non-equivariant baselines, by grouping them based on VIN or GPPN. The results are shown in Figure 8.

**Fixed $K$.** For fixed $K$ setup in Figure 8 (left), we keep number of iterations to be $K = 30$ and kernel size $F = 3$ for all methods.

For SymVIN, it far surpasses VIN for all sizes and preserves the gap throughout the evaluation. Additionally, SymVIN has slightly higher variance across three random seeds (three separately trained models).

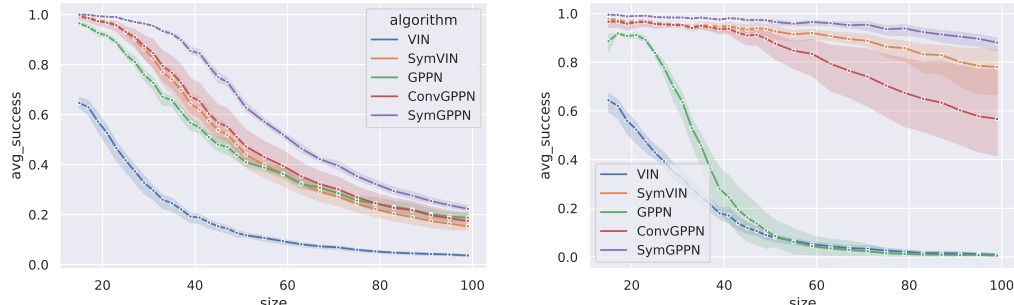

Figure 8: Results for generalization on larger maps for all methods. **(Left)** Fixed $K = 30$ iterations. **(Left)** Variable $K$ iterations, where $K = \sqrt{2} \cdot M$ and $M$ is the generalization map size (x-axis).

Among GPPN and its variants, SymGPPN significantly outperforms both GPPN and ConvGPPN. Interestingly, ConvGPPN has sharper drop with map size than both SymGPPN and ConvGPPN and thus has increasingly larger gap with SymGPPN and finally even got surpassed by GPPN. Across random seeds, the three trained models of ConvGPPN give unexpectedly high variance compared to GPPN and SymGPPN.

**Variable $K$.** We also experiment all methods in the same setting but with variable number of iterations $K = \sqrt{2} \cdot M$, where $M$ is the generalization map size (x-axis). The trend is very different from fixed $K$ setup.

SymVIN generalizes extremely well compared to VIN, although the variance is greater. GPPN seems to diverge for larger variable $K$ since it is even worse than fixed $K = 30$ in all map sizes. ConvGPPN somehow helps convergence, while it fluctuates for different seeds, and SymGPPN is even better and more stable. Surprisingly, SymVIN is even better than ConvGPPN, although injecting symmetry (into SymVIN) does not explicitly deal with convergence.

## C  A Guide to Symmetric Planning

To address the common concern on the accessibility issue for technical section, as a step to solve this, we write a section on explaining the SymVIN method with PyTorch-style pseudocode, since it directly corresponds to what we propose in Section 4 and 5. We try to relate (1) existing concepts with VIN, (2) what we propose in Section 4 and 5 for SymVIN, and (3) actual PyTorch implementation of VIN and SymVIN aligned line-by-line based on semantic correspondence.

We will consider to have another short section on intuitively explaining our Symmetric Planning framework and practical considerations in the next few days.

### C.1  PyTorch-style pseudocode

We provide the key Python code snippets to demonstrate how easy it is to implement SymVIN, our symmetric version of VIN [17].

In the current Section 5 (SymPlan practice), we heavily use the concepts from Steerable CNNs. Thanks to the equivariant network community and the `e2cnn` package, the actual implementation is compact and closely corresponds to their non-equivariant counterpart, VIN, line-by-line. Thus, the ultimate goal here is to illustrate that, whatever concepts we have in regular CNNs (e.g., have whatever channels we want), we can can use steerable CNNs that incorporate desired extra symmetry (of $D_4$ rotation+reflection or $C_4$ rotation).

We highlight the implementation of the value iteration procedure in VIN and SymVIN:

$$V := \max_a R^a + \gamma \times P^a * V. \tag{5}$$

Note that we use actual code snippets to avoid hiding any details.

```
1  import torch
2
3
4
5
6
7
8
9
10
11
12  # Define regular 2D convolution
13  q_conv = torch.nn.Conv2d(
14      in_channels=1,
15      out_channels=2 * q_size,
16      kernel_size=F, stride=1, bias=False
17  )
```

Listing 1: Define 'expected value' convolution layer for VIN.

```
1  import torch
2  import e2cnn
3
4  # Define the symmetry group to be D4
5  gspace = e2cnn.gspaces.FlipRot2dOnR2(N=4)
6  # Define feature (fiber) representations
7  field_type_q_in = e2cnn.nn.FieldType(
8      gspace=gspace,
9      representations=2 * q_size * [gspace.
         regular_repr]
10  )
11  # Define steerable convolution
12  q_r2conv = e2cnn.nn.R2Conv(
13      in_type=field_type_q_in,
14      out_type=field_type_q_out,
15      kernel_size=F, stride=1, bias=False
16  )
```

Listing 2: Define 'expected value' (steerable) convolution layer for SymVIN.

**Defining (steerable) convolution layer.** First, we show the definition of the key convolution layer for a key operation in VIN and SymVIN: expected value operator, in Listing 1 and 2.

As proved in Theorem 4.2, the expected value operator can be executed by a steerable convolution layer for (2D) path planning. This serves as the theoretical foundation on how we should use a steerable layer here.

For the left side, a regular 2D convolution is defined for VIN. The right side defines a steerable convolution layer, using the library e2cnn from [16]. It provides high-level abstraction for building equivariant 2D steerable convolution networks. As a user, we only need to specify how the feature fields transform (as shown in Figure 3), and it will solve the $G$-steerability constraints, process what needs to be trained for equivariant layers, etc. We use name q_r2conv to highlight the difference.

**Value iteration procedure.** Second, we compare the for loop for value iteration updates in VIN and SymVIN, where the former one has regular 2D convolution Conv2D (Listing 3), and the latter one uses steerable convolution [16] (Listing 4).

The lines are aligned based on semantic correspondence. The e2cnn layers, including steerable convolution layers, operate on its GeometricTensor data structure, which is to wrap a PyTorch tensor. We denote them with _geo suffix. It only additionally needs to specify how this tensor (feature field) transforms under a group (e.g., $D_4$), i.e. the user needs to specify a group representation for it.

tensor_directsum is used to concatenate two GeometricTensor's (feature fields) and compute their associated representations (by direct-sum).

Thus, the e2cnn steerable convolution layer on the right side q_r2conv can be used as a regular PyTorch layer, while the input and output are GeometricTensor.

We also define the $\max$ operation as a customized max-pooling layer, named q_max_pool. The implementation is similar to the left side of VIN and needs to additionally guarantee equivariance, and the detail is omitted.

Note that for readability, we assume we use regular representations for the Q-value field $Q$ and the state-value field $V$. They are empirically found to work the best. This corresponds to the definition in field_type_q_in in line 9 in the SymVIN definition listing and the comments in line 16-17 in the steerable VI procedure listing for SymVIN.

Other components are omitted.

# D  Simplified Version: Symmetric Planning

This is a new preliminary version during the rebuttal period that aims to introduce symmetric planning in a more intuitive way, with minimum prerequisites of equivariant networks. This section is intended to be an alternative and more intuitive version to Section 4 (SymPlan framework) and Sec-

```
1  # Input: maze and goal map, #iterations K
2
3
4
5
6  x = torch.cat([maze_map, goal_map], dim=1)
7
8  r = r_conv(x)
9
10 # Init value function V
11 v = torch.zeros(r.size())
12
13
14 for _ in range(K):
15     # Concat and convolve V with P
16     rv = torch.cat([r, v], dim=1)
17     q = q_conv(rv)
18
19     # Max over action channel
20     # > Q: batch_size x q_size x W x H
21     # > V: batch_size x 1 x W x H
22     q = q.view(-1, q_size, W, H)
23     v, _ = torch.max(q, dim=1)
24     v = v.view(-1, W, H)
25
26 # Output: 'q' (to produce policy map)
```

```
1  # Input: maze and goal map, #iterations K
2
3  from e2cnn.nn import GeometricTensor
4  from e2cnn.nn import tensor_directsum
5
6  x = torch.cat([maze_map, goal_map], dim=1)
7  x_geo = GeometricTensor(x, type=field_type_x)
8  r_geo = r_r2conv(x_geo)
9
10 # Init V and wrap V in e2cnn 'geometric tensor'
11 v_raw = torch.zeros(r_geo.size())
12 v_geo = GeometricTensor(v_raw, field_type_v)
13
14 for _ in range(K):
15     # Concat (direct-sum) and convolve V with P
16     rv_geo = tensor_directsum([r_geo, v_geo])
17     q_geo = q_r2conv(rv_geo)
18
19     # Max over group channel
20     # > Q: batch_size x (|G| * q_size) x W x H
21     # > V: batch_size x (|G| * 1) x W x H
22     v_geo = q_max_pool(q_geo)
23
24
25
26 # Output: 'q_geo' (to produce policy map)
```

Listing 3: The central value iteration procedure for VIN. Some variable names are adjusted accordingly for readability. W and H are width and height for 2D map.

Listing 4: The equivariant steerable value iteration procedure for SymVIN. Lines are aligned by semantic correspondence. Definition of other field types are similar and thus omitted.

tion 5 (SymPlan in practice). We would appreciate feedback and consider to make further revision to this section and the organization of the entire paper.

## D.1  Overview

In this work, we aim to exploit the inherent symmetry in a broadly existed problem: path planning. Intuitively, since a rotated or reflected 2D map are still another instance of 2D map, such as in Figure 1, their policies and optimal paths are related. This unveils an inherent symmetry property of the path planning problem on the 2D grid that we could exploit.

In our work, we provide a rigorous algorithmic framework that can *provably* make use of symmetry in an *efficient* manner. In this section, we will first introduce the algorithm we are based on: Value Iteration Networks (VINs) [17], and use it as foundation to build our algorithm: Symmetric VIN. Finally, we provide intuition to the theoretical guarantees on how we make use of symmetry.

## D.2  Value Iteration Network: Background and Interpretation

Value Iteration Network (VIN) [17] is an example of a differentiable planning algorithm. It empirically found that, for 2D path planning, value iteration can be implemented by a deep convolution network.

**Background: VIN.**  Value iteration is an instance of a dynamic programming (DP) method to solve Markov decision processes (MDPs). It iteratively applies the Bellman (optimality) operator until convergence, which is based on the following Bellman (optimality) equation:

$$Q(s, a) = R(s, a) + \gamma \sum_{s'} P(s'|s, a)V(s'), \quad V(s) = \max_a Q(s, a) \tag{6}$$

Tamar et al. [17] used a convolution network to parameterize value iteration. It jointly learns in a latent MDP on 2D grid, which has the latent reward function $\bar{R} : \mathbb{Z}^2 \to \mathbb{R}^{|\mathcal{A}|}$ and value function $\bar{V} : \mathbb{Z}^2 \to \mathbb{R}$, and applies value iteration on that MDP:

$$\bar{Q}_{\bar{a},i',j'}^{(k)} = \bar{R}_{\bar{a},i,j} + \sum_{i,j} W_{\bar{a},i,j}^V \bar{V}_{i'-i,j'-j}^{(k-1)} = \bar{R}_{\bar{a},i,j} + \texttt{Conv2D}(\bar{V}; W^V), \quad \bar{V}_{i,j}^{(k)} = \max_{\bar{a}} \bar{Q}_{\bar{a},i',j'}^{(k)} \tag{7}$$

Later, we generalize the idea of VIN that (1) represents reward and value functions as fields on 2D grid, and (2) realizes value iteration by operations on the fields. Our final goal is to use VIN

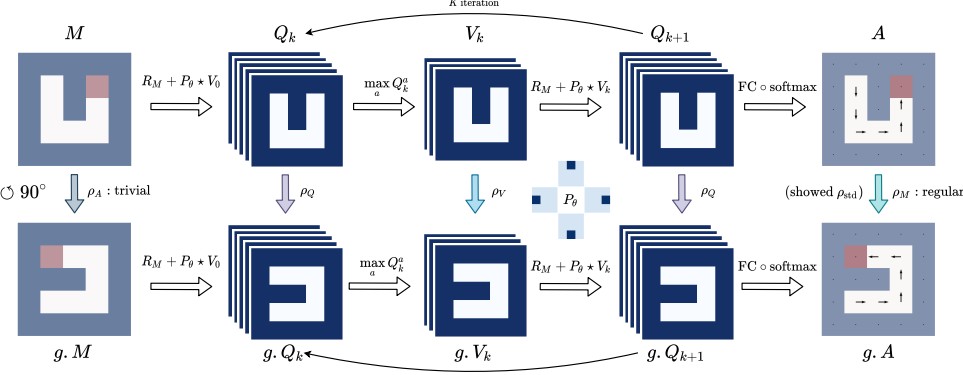

Figure 9: (This is a copy of Figure 3.) The commutative diagram of Symmetric Value Iteration Network (SymVIN). Every *row* is a full computation graph of VIN. Every *column* is to rotate field by $\circlearrowleft 90°$. The key message: if we *rotate* the map (from $M$ to $g.M$), to guarantee the final policy function to also be *equivalently rotated* (from $A$ to $g.A$), we shall guarantee every *transformation* (e.g., $Q_k \mapsto V_k$ and $V_k \mapsto Q_{k+1}$) in value iteration to also be *equivariant* ($g.f(x) = f(g.x)$, for every *pair of columns*).

to demonstrate a principled method for incorporating symmetry in differentiable planning. After reviewing the basics of VIN, we will next summarize the reasoning of choosing VIN.

### D.3 Symmetric Value Iteration Network: A Practical Symmetric Planning Algorithm

**Why do we choose VIN?** There are two reasons behind the choice of VIN.

1. The expected value operator in value iteration $\sum_{s'} P(s'|s,a)V(s')$ is *linear* in value function. As we show in Theorem 4.1, it is also *equivariant* for (2D) path planning, this means that it is a *linear equivariant operator*. According to Cohen et al. [12], any linear equivariant operator (on spaces such as 2D grid) has one-to-one correspondence to a (group equivariant) convolution operator.

2. Value iteration, or Bellman (optimality) operator, is fully convolutional, i.e. only relies on operating on functions ("fields") over $\mathbb{Z}^2$, such as value function, reward function, and transition functions: $V_{k+1}(s) = \max_a R^a(s) + \gamma \times [P^a \star V_k](s)$. This enables us to inject symmetry by enforcing equivariance within convolution. For example, for a 2D map in Figure 1, the 4 corner states are symmetric under any one of the eight transformations in $D_4$, and we can enforce those 4 states to have the same value if we rotate or flip the map ($D_4$-equivariance). This avoids the need to find if a new state is symmetric to any existing state, which is shown to be NP-hard [8].

In summary, VIN satisfies both desiderata: (1) it uses convolution as the backbone, and (2) it operates on fields. Furthermore, we find VIN is empirically and conceptually the *simplest* differentiable planning algorithm that satisfies them, which leads to our decision.

**How to inject symmetry?** VIN uses a regular 2D convolutional network (Equation 7), which has *translation equivariance* [15, 13]. More concretely, a VIN will output the same value function for the same map patches that only differ by 2D translation. We omit how to characterize translation equivariance here, since it requires a different mechanism to handle and does not *decrease* the search space nor *reduce* a path planning MDP to an easier problem.

Beyond translation, we are more interested in *rotation* and *reflection* symmetries. Intuitively, as in Figure 1, if we find the optimal solution to a map, it automatically **generalizes** the solution to all 8 transformed maps (4 rotations times 2 reflections, including identity transformation). This can be characterized by *equivariance* of a planning algorithm Plan, such as value iteration VI, visualized in Figure 9: $g.\text{Plan}(M) = \text{Plan}(g.M)$, where $M$ is a maze map, and $g$ is the symmetry group $D_4$ under which 2D grid is invariant.

More importantly, symmetry also helps **training** of differentiable planning algorithms. Intuitively, symmetry in path planning poses additional constraint to its search space: if the goal is in the north,

go up; if in the east, go right. In other words, the knowledge can be shared between symmetric cases, or the path planning is effectively reduced by symmetry to a smaller one. This property can also be depicted by equivariance of Bellman operators $\mathcal{T}$, or a step of value iteration: $g.\mathcal{T}[V_0] = \mathcal{T}[g.V_0]$. If we use $\mathtt{VI}(M)$ to denote applying Bellman operators on arbitrary initialization until convergence $\mathcal{T}^\infty[V_0]$, value iteration is also equivariant, as demonstrated in Figure 9:

$$g.\mathtt{VI}(M) \equiv g.\mathcal{T}^\infty[V_0] = \mathcal{T}^\infty[g.V_0] \equiv \mathtt{VI}(g.M). \tag{8}$$

Thus, we inject equivariance into value iteration w.r.t. *rotation* and *reflection*, in addition to *translation*, through **steerable convolution (network)** from Cohen and Welling [14], which exactly matches our criteria. Cohen et al. [12] prove that steerable convolution is the most general linear equivariant map under some conditions, which value iteration satisfies. Weiler and Cesa [16] build $E(2)$-Steerable CNNs for 2D space, and we use their package e2cnn in our implementation. In practice, to inject symmetry into VIN, we simply need to replace the translation-equivariant Conv2D with SteerableConv:

$$\bar{Q}_{\bar{a},i',j'}^{(k)} = \bar{R}_{\bar{a},i,j} + \mathtt{SteerableConv}(\bar{V}; W^V), \quad \bar{V}_{i,j}^{(k)} = \max_{\bar{a}} \bar{Q}_{\bar{a},i',j'}^{(k)}. \tag{9}$$

We formally justify our design in Section D.4 below and provide more technical details in Section 4.

## D.4 Theoretical Justification: Why does it work?

In the Section D.3, we show how to exploit symmetry in path planning by equivariance from convolution via intuition. The goal of this new section is to (1) connect the theoretical justification with the algorithmic design, and (2) provide intuition for the justification. Even through we focus on a specific task, we hope that the underlying guidelines on integrating symmetry into planning are useful for broader planning algorithms and problems as well.

This version is for the purposes of rebuttal and preview, so we may refer some details to the original Section 4. We will consider further revision depending on how to form the method section.

**Overview.** There are numerous types of symmetry in various planning tasks. We study symmetry in **path planning** as an example, because it is a straightforward planning problem, and its solutions have been intensively studied in robotics and artificial intelligence [53, 1]. However, even for this problem, the symmetry has *not* been *effectively* exploited in its planning algorithms, such as Dijkstra's algorithm, A*, or RRT, because of NP-hard orbit finding [8]. Additionally, we focus on **value iteration** because it is both widely use and connects closely with convolution [14].

**Theory: symmetry in planning.** If we want to exploit symmetry in a task to improve planning, there are two major steps: (1) characterize the symmetry in the task, and (2) incorporate corresponding symmetry into the planning algorithm. The theoretical results in Section 4.2 mainly characterize the symmetry and direct us to a feasible planning algorithm.

The **symmetry in tasks** or MDPs can be specified by the equivariance property of the transition and reward function, studied in Ravindran and Barto [3], van der Pol et al. [32]:

$$\bar{P}(s' \mid s, a) = \bar{P}(g.s' \mid g.s, g.a), \quad \forall g \in G, \forall s, a, s' \tag{10}$$

$$\bar{R}_M(s, a) = \bar{R}_{g.M}(g.s, g.a), \quad \forall g \in G, \forall s, a \tag{11}$$

Note that how the group $G$ acts on states and actions is decided by the space $\mathcal{S}$ or $\mathcal{A}$, which has been discussed in Equation 1 in Section 4.2. We emphasize that the equivariance property of the reward function is different from prior work [3, 32]: in our case, the reward function encodes obstacles as well, and thus depends on map input $M$. Intuitively, using Figure 1 as an example, if a position $s$ is rotated $g.s$, to find how the correct original reward $R$, the input map $M$ must also be rotated $g.M$. More details in Section 4.2 and Section H.

As for exploiting the **symmetry in planning algorithms**, we focus on value iteration and the VIN algorithm. We first prove in Theorem 4.1 that value iteration for path planning respects the *equivariance* property. This confirms that value iteration is a feasible method to incorporate symmetry. The next result in Theorem 4.2 further proves that value iteration is a general form of convolution (*steerable convolution*), motivating the use of steerable CNNs by Cohen and Welling [14] to replace regular CNNs in VIN.

X

**Retrospect.** We study how to inject symmetry into VIN for (2D) path planning, and expect the task-specific technical details are useful for two types of readers. *(i) Using VIN.* If one uses VIN for differentiable planning, the resulting algorithms SymVIN or SymGPPN can be a plug-in alternative, as a part in a larger end-to-end system. *(ii) Studying path planning.* The proposed framework characterizes the symmetry in (2D) path planning, so it is possible to apply the underlying ideas to other domains. For example, it is possible to extend to higher-dimensional continuous Euclidean spaces.

This concludes the section. We appreciate any feedback for this new simplified section on implementation and theory of symmetric planning. We will keep improving it and better integrate with the current "detailed" version in the future iterations.

## E   Experiments: Details and Additional Results (moved)

### E.1   Details: Setup

**Action space.**   Note that the MDP action space $\mathcal{A}$ needs to be *compatible* with the group action $G \times \mathcal{A} \to \mathcal{A}$. Since the E2CNN package [16] uses *counterclockwise* rotations as generators for rotation groups $C_n$, the action space needs to be *counterclockwise*.

**Mapper training: manipulation.**   During training, we pre-train the mapper and the planner separately for 15 epochs. Where the mapper takes manipulator workspace and outputs configuration space. The mapper is trained to minimize the binary cross entropy between output and ground truth configurations space. The planner is trained in the same way as described in Section 6.1. After pre-training, we switch the input to the planner from ground truth configuration space to the one from the mapper. During testing, we follow the pipeline in [37] that the mapper-planner only have access to the manipulator workspace.

### E.2   Details: Environments.

**Manipulation.**   For planning in configuration space, the configuration space of the 2 DoFs manipulator has no constraints in the $\{0, \pi\}$ boundaries, i.e., no joint limits. To reflect this nature of the configuration space in manipulation tasks, we use circular padding before convolution operation. The circular padding is applied to convolution layers in VIN, SymVIN, ConvGPPN, and SymGPPN. Moreover, in GPPN, there is a convolution encoder before the LSTM layer. We add the circular padding in the convolution layers in GPPN as well.

In **2-DOF manipulation** in configuration space, we adopt the setting in [37] and train networks to take as input of configuration space, represented by two joints. We randomly generate 0 to 5 obstacles in the manipulator workspace. Then the 2 degree-of-freedom (DOF) configuration space is constructed from workspace and discretized into 2D grid with sizes $\{18, 36\}$, corresponding to bins of $20°$ and $10°$, respectively.

We allow each joint to rotate over $2\pi$, so the configuration space of 2-DOF manipulation forms a torus $\mathbb{T}^2$. Thus, the both boundaries need to be connected when generating action demonstrations, and (equivariant) convolutions need to be circular (with padding mode) to wrap around for all methods. We allow each joint to rotate over $2\pi$, so the both boundaries in configuration space need to be connected when generating action demonstrations, and (equivariant) convolutions need to be circular (with padding mode) to wrap around for all methods.

### E.3   Details: Model Architecture

We try to mimic the setup in VIN and GPPN [18].

For non-SymPlan related parameters, we use learning rate of $10^{-3}$, batch size of 32 if possible (GPPN variants need smaller), RMSprop optimizer.

For SymPlan parameters, we use 150 hidden channels (or 150 *trivial* representations for SymPlan methods) to process the input map. We use 100 hidden channels for Q-value for VIN (or 100 *regular* representations for SymVIN), and use 40 hidden channels for Q-value for GPPN and ConvGPPN

(or 40 *regular* representations for SymGPPN on $15 \times 15$, and 20 for larger maps because of memory constraint).

## E.4  Visualization of learned models

We visualize a trained VIN and a SymVIN, evaluated on a $15 \times 15$ map and its rotated version. For non-symmetric VIN in Figure 10, the learned policy is obviously not equivariant under rotation.

We also visualize SymVIN on larger map sizes: $28 \times 28$ and $50 \times 50$, to demonstrate its performance and equivariance.

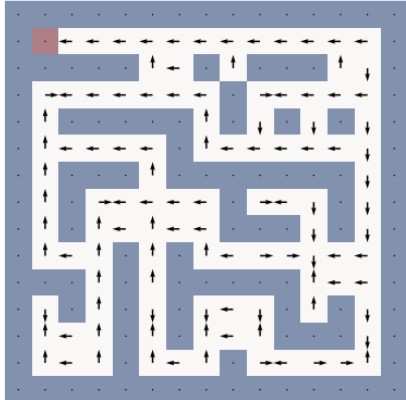 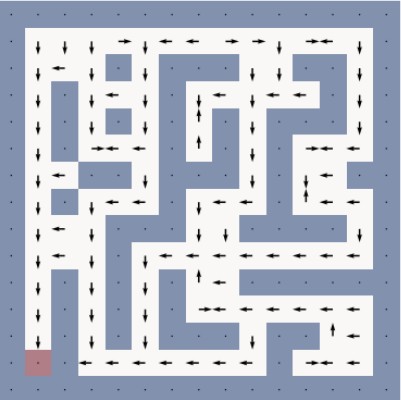

Figure 10: A trained VIN evaluated on a $15 \times 15$ map and its rotated version. It is obvious that the learned policy is not equivariant under rotation.

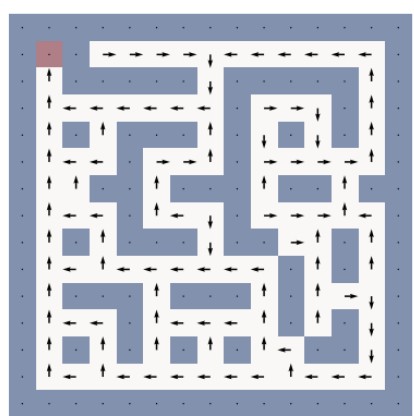 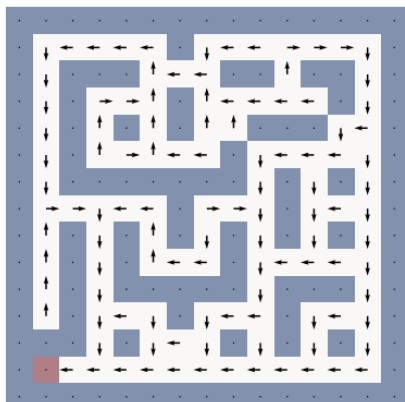

Figure 11: A trained SymVIN evaluated on a $15 \times 15$ map and its rotated version.

## E.5  Further Analysis

**Additional training curves.**    We also provide other training curves that we only show test numbers in the main text.

**Training efficiency with less data.**    Since the supervision is still dense, we experiment on training with even smaller dataset to experiment in more extreme setup. We experiment how symmetry may affect the training efficiency of Symmetric Planners by further reducing the size of training dataset. We compare on two environments: 2D navigation and visual navigation, with training/validation/test size of 1K/200/200, for all methods.

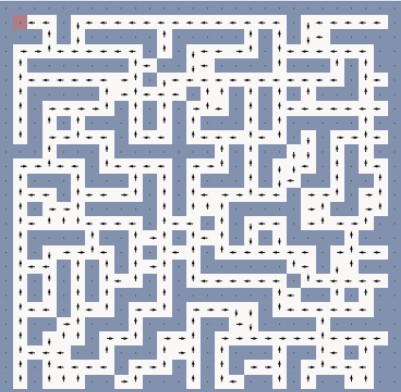 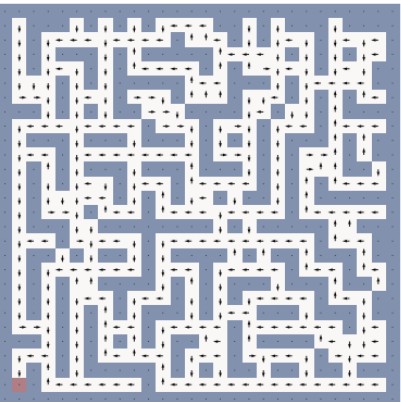

Figure 12: A fully trained SymVIN evaluated on a $28 \times 28$ map and its rotated version.

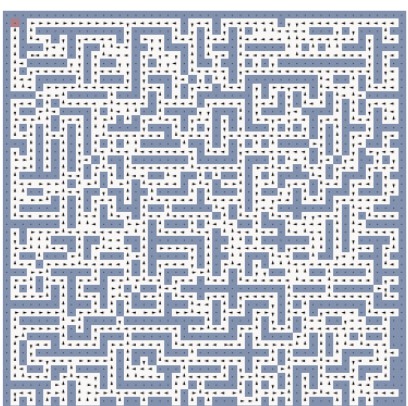 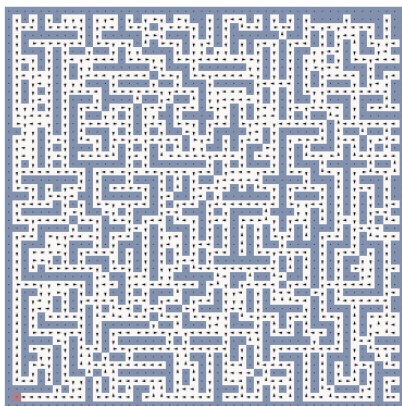

Figure 13: A fully trained SymVIN evaluated on a $50 \times 50$ map and its rotated version.

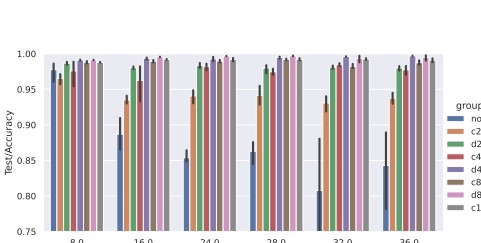 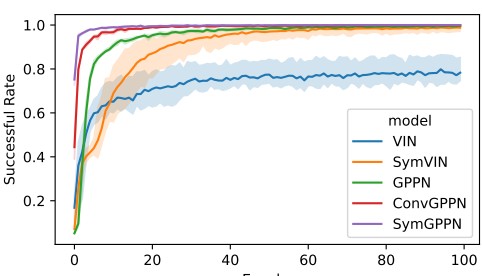

Figure 14: **(Left)** Accuracy evaluated on unseen test maps. The x-axis is the width of the map, and the y-axis is the accuracy, reported on every map size and every size and every chose symmetry group $G$. **(Right)** Visual navigation $15 \times 15$ with 10K data.

**Choose of symmetry groups for navigation.** One important benefit of partially equivariant network is that, we do not need to design the group representation of MDP action space $\rho_{\mathcal{A}}(g)$ for different group or action space. Thus, we experiment several $G$-equivariant variants with different group equivariance: (discrete rotation group) $C_2, C_4, C_8, C_{16}$, and (dihedral group) $D_2, D_4, D_8$, all based on $E(2)$-steerable CNN [16]. For all intermediate layers, we use regular representations $\rho_{\mathrm{reg}}(g)$ of each group, followed by a final policy layer with non-equivariant $1 \times 1$ convolution.

The results are reported in the Figure 14 (left). We only compare VIN (denoted as "none" symmetry) against our E(2)-VIN (other symmetry group option) on 2D navigation with $15 \times 15$ maps.

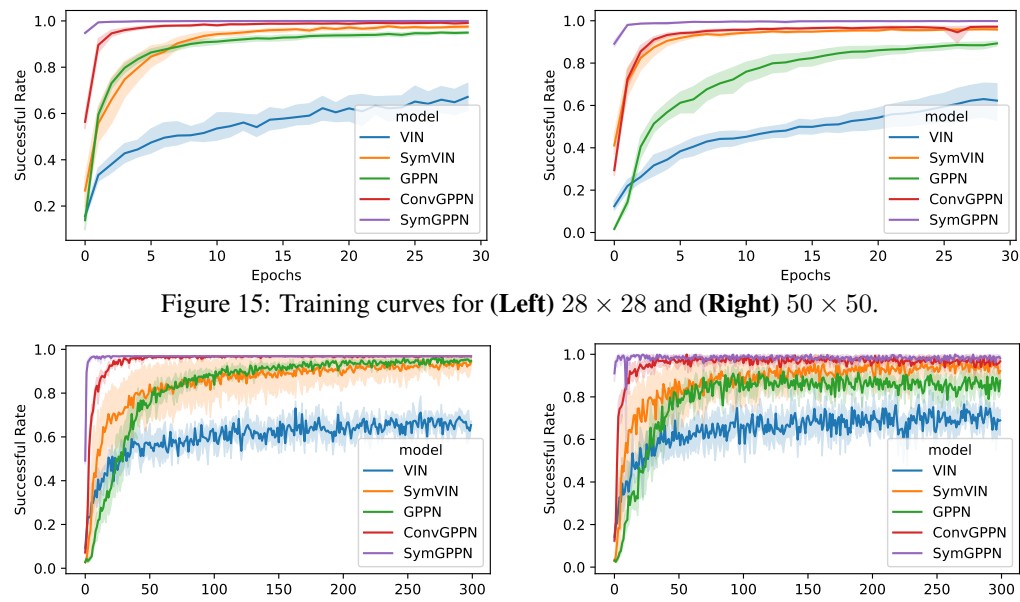

Figure 15: Training curves for **(Left)** $28 \times 28$ and **(Right)** $50 \times 50$.

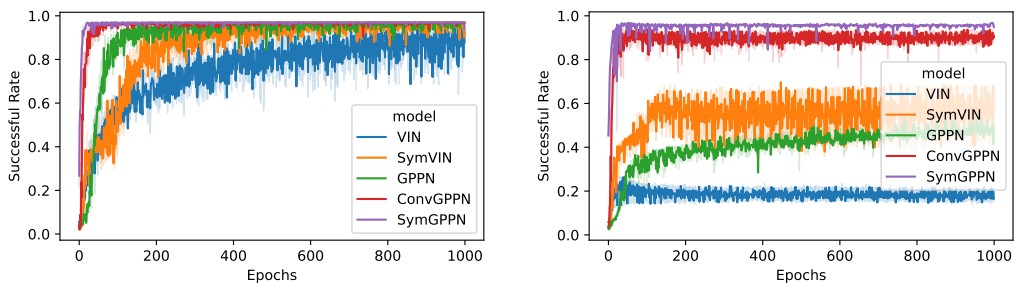

Figure 16: Training curves for $15 \times 15$ 2D navigation 1K data **(Left)** training and **(Right)** validation successful rate.

Figure 17: Training curves for $15 \times 15$ visual navigation 1K data **(Left)** training and **(Right)** validation successful rate.

Table 2: Fiber representations

| (Fiber representation) | SymVIN |
| --- | --- |
| Default | 98.45 |
| Hidden: trivial to regular | 99.07 |
| State-value $\rho_V$: regular to trivial | 63.08 |
| Q-value $\rho_Q$: regular to trivial | 21.30 |
| $\rho_Q$ and $\rho_V$: both trivial | 2.814 |

In general, the planners equipped with any $G$ group equivariance outperform the vanilla non-equivariant VIN, and $D_4$-equivariant steerable CNN performs the best on most map sizes. Additionally, since the environment has actions in 8 directions (4 diagonals), $C_8$ or $D_8$ groups seem to take advantage of that and have slightly higher accuracy on some map sizes, while $C_{16}$ is over-constrained compared to the true symmetry $G = D_4$ and be detrimental to performance. The non-equivariant VIN also experiences higher variance on large maps.

**Choosing fiber representations.** As we use steerable convolutions [16] to build symmetric planners, we are free to choose the representations for feature fields, where intermediate equivariant convolutional layers will be equivariant between them $f(\rho_{\mathrm{in}}(g)x) = \rho_{\mathrm{out}}(g)f(x)$. We found representations for some feature fields are critical to the performance: mainly $V : \mathcal{S} \to \mathbb{R}$ and $Q : \mathcal{S} \to \mathbb{R}^{|A|}$.

We use the best setting as default, and ablate every option. As shown in Table 2, changing $\rho_V$ or $\rho_Q$ to trivial representation would result in much worse results.

**Fully vs. Partially equivariance for symmetric planners.** One seemingly minor but critical design choice in our SymPlan networks is the choice of the final policy layer, which maps Q-values $\mathcal{S} \to \mathbb{R}^{|\mathcal{A}|}$ to policy logits $\mathcal{S} \to \mathbb{R}^{|\mathcal{A}|}$. Fully equivariant is expected to perform better, but it has some points worth to mention. (1) We experience unstable training at the beginning, where the loss can go up to $10^6$ in the first epoch, while we did not observe it in non-equivariant or partially equivariant counterparts. However, this only slightly affects training.

In summary, we found even though fully equivariant version can perform slightly better in the best tuned setting, on average setting, partially equivariant version is more robust and the gap is much larger, as shown in the follow table, which an example of averaging over three choices of representations introduced in the last paragraph. On average partially equivariant version is much better. In our experiments, partially equivariant version also is easier to tune.

Table 3: Fully vs. Partially equivariance

| (Equivariance) | SymVIN |
|---|---|
| *Partially* equivariant averaged over all representations | 91.04 |
| *Fully* equivariant averaged over all representations | 42.61 |

# F    Additional Discussion

## F.1    Limitations and Extensions

**Assumption on known domain structure.** As in VIN, although the framework of steerable planning can potentially handle different domains, one important hidden assumption is that the underlying domain $\Omega$ (state space), is known. In other words, we fix the structure of learned transition kernels $p(s' \mid s, a)$ and estimate coefficients of it. One potential method is to use Transformers that learn attention weights to all states in $\mathcal{S}$, which has been partially explored in SPT [37]. Additionally, it is also possible to treat unknown MDPs as learned transition graphs, as explored in XLVIN [38]. We leave the consideration of symmetry in unknown underlying domains for future work.

**The curse of dimensionality.** The paradigm of steerable planning still requires full expansion in computing value iteration (opposite to *sampling-based*), since we realize the symmetric planner using group equivariant convolutions (essentially summation or integral). Convolutions on high-dimensional space could suffer from the curse of dimensionality for higher dimensional domains, and are vastly under-explored. This is a primary reason why we need sampling-based planning algorithms. If the domain (state-action transition graph) is sparsely connected, value iteration can still scale up to higher dimensions. It is also unclear either when steerable planning would fail, or how sampling-based algorithms could be integrated with the symmetric planning paradigm.

## F.2    The considered symmetry in spatial MDPs

We need to differentiate between two types of symmetry in MDPs. Let's take spatial graph as illustrative example to understand the potential symmetry from a higher level, which means that the nodes $\mathcal{V}$ in the graph have spatial coordinates $\mathbb{Z}^n$ or $\mathbb{R}^n$. Our 2D path planning is a special case of spatial graph, where the actions can only move to adjacent spatial nodes.

Let the graph denoted as $\mathcal{G} = \langle \mathcal{V}, \mathcal{E} \rangle$. $\mathcal{E}$ is the set of edges connecting two states with an action. One type of symmetry is the symmetry of the graph itself. For the grid case, it means that after $D_4$ rotation or reflection, the map is unchanged.

Another type of symmetry comes from the isometries of the space. For a spatial graph, we can rotate it freely in a space, while the relative positions are unchanged. For our grid case, it is shown in the Figure 1 that rotating a map resulting in the rotated policy. However, the map or policy itself can never be equal under any transformation in $D_4$.

In other words, the first type is symmetry within a MDP (rely on the property of the MDP itself $\mathcal{M}$, or $\mathrm{Aut}(\mathcal{M})$), and the second type is symmetry between MDPs (only rely on the property of the underlying spatial space $\mathbb{Z}^2$, or $\mathrm{Aut}(\mathbb{Z}^2)$).

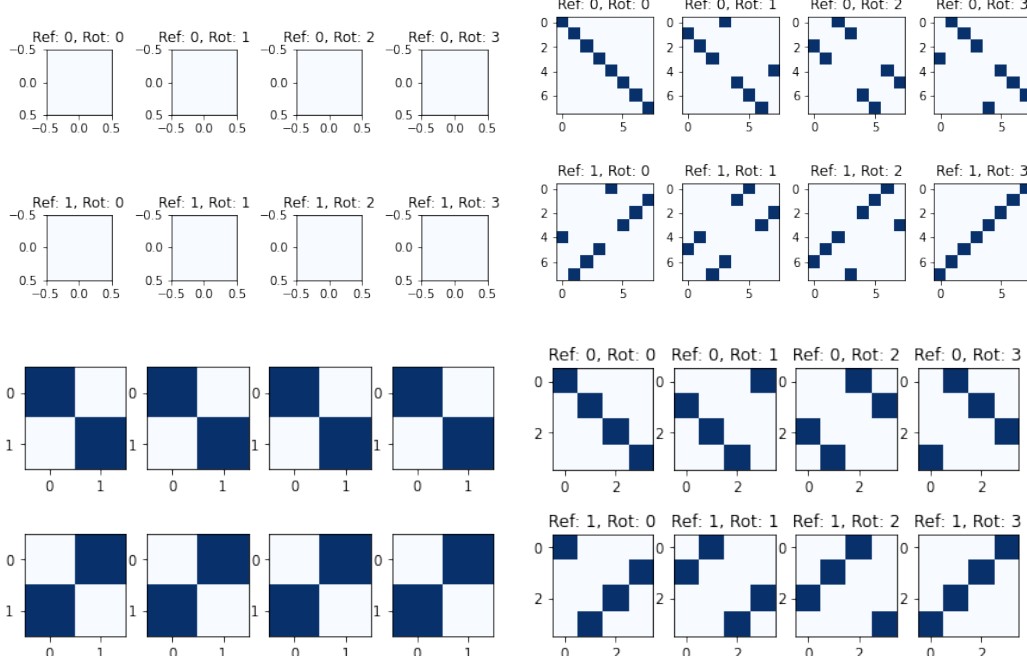

Figure 18: Visualization of the permutation representations of $D_4$ group for every element $g \in D_4$ (4 rotations each row and 2 reflections each column). They are (1) the trivial representation, (2) the regular representation, (3) the quotient representation (quotienting out *rotations*), (4) the quotient representation (quotienting out *reflections*).

Nevertheless, we could input map $M$ and somehow treat symmetric states between MDPs as one state. See the proofs section for more details.

## G  Additional Background and Concepts

### G.1  Group representations: visual understanding

A group representation is a (linear) group action that defines how a group acts on some space. Cohen and Welling [15, 14], Weiler and Cesa [16] provide more formal introduction to them in the context of equivariant neural networks. We provide visual understanding and refer the readers to them for comprehensive account.

To visually understand how the group $D_4$ acts on some vector space, we visualize the trivial, regular, and quotient (quotienting out reflections $sr^2$) representations, which are *permutation matrices*. If we apply such a representation $\rho(g)(g \in D_4)$ to a vector, the elements get *cyclically permuted*. See Figure 18.

The quotient representation that quotients out reflections and has dimension $4 \times 4$ is what we need to use on the 4-direction action space.

### G.2  Geometric Deep Learning

We review another set of important concepts that motivate our formulation of steerable planning: geometric deep learning and the theories on connecting equivariance and convolution [11, 12, 13]. Bronstein et al. [11] use $x$ for feature fields while Cohen and Welling [14], Cohen et al. [12], Weiler and Cesa [16] use $f$.

**Convolutional feature fields.**  The signals are taken from set $\mathcal{C} = \mathbb{R}^D$ on some structured *domain* $\Omega$, and all mappings from the domain to signals forms the space of $\mathcal{C}$-valued signals $\mathcal{X}(\Omega, \mathcal{C}) = \{f : \Omega \to \mathcal{C}\}$, or $\mathcal{X}(\Omega)$ for abbreviation. For instance, for RGB images, the domain is the 2D $n \times n$

grid $\Omega = \mathbb{Z}_n \times \mathbb{Z}_n$, and every pixel can take RGB values $\mathcal{C} = \mathbb{R}^3$ at each point in the domain $u \in \Omega$, represented by a mapping $x : \mathbb{Z}_n \times \mathbb{Z}_n \to \mathbb{R}^3$. A function on images thus operates on $3n^2$-dimensional inputs.

It is argued that the underlying geometric structure of domains $\Omega$ plays key role in alleviating the curse of dimensionality, such as convolution networks in computer vision, and this framework is named *Geometric Deep Learning*. We refer the readers to Geometric Deep Learning [11] for more details, and to more rigorous theories on the relation between equivariant maps and convolutions in [12] (vector fields through induced representations) and [13] (scalar fields through trivial representations).

**Group convolution.** Convolutions are shift-equivariant operations, and vice versa. This is the special case for $\Omega = \mathbb{R}$, which can be generalized to any group $G$ (that we can integrate or sum over). The *group convolution* for signals on $\Omega$ is then defined[5] as

$$(f \star \psi)(g) = \langle f, \rho(g)\psi \rangle = \int_\Omega f(u)\psi(g^{-1}u)\mathrm{d}u, \tag{12}$$

where $\psi(u)$ is shifted copies of a filter, usually locally supported on a subset of $\Omega$ and padded outside. Note that although $x$ takes $u \in \Omega$, the feature map $(x \star \psi)$ takes as input the elements $g \in G$ instead of points on the domain $u \in \Omega$. All following group convolution layers take $G$: $\mathcal{X}(G) \to \mathcal{X}(G)$. In the grid case, the domain $\Omega$ is *homogeneous* space of the group $G$, i.e. the group $G$ acts transitively: for any two points $u, v \in \Omega$ there exists a symmetry $g \in G$ to reach $u = gv$.

Analogous to classic shift-equivariant convolutions, the generalized group convolution is $G$-equivariant [12]. It is observed that $\langle x, \rho(g)\theta \rangle = \langle \rho(g^{-1})x, \theta \rangle$, and from the defining property of group representations $\rho(h^{-1})\rho(g) = \rho(h^{-1}g)$, the $G$-equivariance of group convolution follows [11]:

$$(\rho(h)x \star \theta)(g) = \langle \rho(h)x, \rho(g)\theta \rangle = \langle x, \rho(h^{-1}g)\theta \rangle = \rho(h)(x \star \theta)(g) \tag{13}$$

**Steerable convolution kernels.** Steerable convolutions extend group convolutions to more general setup and decouple the computation cost with the group size [14, 56]. For example, $E(2)$-steerable CNNs [16] apply it for $E(2)$ group, which is semi-direct product of translations $\mathbb{R}^2$ and a fiber group $H$, where $H$ is a group of transformations that fixes the origin and is $O(2)$ or its subgroups. The representation on the signals/fields is induced from a representation of the fiber group $H$. Use $\mathbb{R}^2$ as example, a steerable kernel only needs to be $H$-equivariant by satisfying the following constraint [16]:

$$\psi(hx) = \rho_{\text{out}}(h)\psi(x)\rho_{\text{in}}(h^{-1}) \quad \forall h \in H, x \in \mathbb{R}^2. \tag{14}$$

### G.3 Steerable CNNs

We still use the running example on $\mathbb{Z}^2$ and group $p4m = \mathbb{Z}^2 \rtimes D_4$.

**Induced representations.** We follow [14, 12] to use $\pi$ for *induced* representations. We still use feature fields over $\mathbb{Z}^2$ as example.

As shown in **Figure 2 middle**, to transform a feature field $f : \mathbb{Z}^2 \to \mathbb{R}^C$ on base $\mathbb{Z}^2$ with group $p4m = \mathbb{Z}^2 \rtimes D_4$, we need the *induced representation* [14, 12]. The induced representation in this case is denoted as $\pi(g) \triangleq \text{ind}_{D_4}^{\mathbb{Z}^2 \rtimes D_4} \rho(g)$ (for all $g$), which means how the group action of $D_4$ transforms a feature field on $\mathbb{Z}^2 \rtimes D_4$.

It acts on the feature field with two parts: (1) on the base space $\mathbb{Z}^2$ and (2) on the fibers (feature channels $\mathbb{R}^C$) by fiber group $H = D_4$ [14, 16]. More specifically, applying a translation $t \in \mathbb{Z}^2$ and a transformation $r \in D_4$ to some field $f$, we get $\pi(tr)f$ [14, 16]:

$$f(x) \mapsto [\pi(tr)f](x) \triangleq \rho(r) \cdot \left[ f\left((tr)^{-1}x\right) \right]. \tag{15}$$

---

[5]The definition of group convolution needs to assume that (1) signals $\mathcal{X}(\Omega)$ are in a Hilbert space (to define an inner product $\langle x, \theta \rangle = \int_\Omega x(u)\theta(u)\mathrm{d}u$) and (2) the group $G$ is locally compact (so a Haar measure exists and "shift" of filter can be defined).

$\rho(r)$ is the fiber representation that transforms the fibers $\mathbb{R}^C$, and $(tr)^{-1}x$ finds the element before group action (or equivalently transforming the base space $\mathbb{Z}^2$). Thus, $\pi$ only depends on the fiber representation $\rho$ but not the latter part, thus named *induced representation* by $\rho$.

**Steerable convolution vs. group convolution.** The steerable convolution on $\mathbb{Z}^2$ The understanding of this point helps to understand how a group acts on various feature fields and the design of state space for path planning problems. We use the discrete group $p4 = \mathbb{Z}^2 \rtimes C_4$ as example, which consists of $\mathbb{Z}^2$ translations and $90°$ rotations. The only difference with $p4m$ is $p4$ does not have reflections.

The group convolution with filter $\psi$ and signal $x$ on grid (or $\mathbf{p} \in \mathbb{Z}^2$), which outputs signals (a function) on group $p4$

$$[\psi \star x](\mathbf{t}, r) := \sum_{\mathbf{p} \in \mathbb{Z}^2} \psi((\mathbf{t}, r)^{-1}\mathbf{p}) \, x(\mathbf{p}). \tag{16}$$

A group $G$ has a natural action on the functions over its elements; if $x : G \to \mathbb{R}$ and $g \in G$, the function $g.x$ is defined as $[g.x](h) := x(g^{-1} \cdot h)$.

For example: The group action of a rotation $r \in C_4$ on the space of functions over $p4$ is

$$[r.y](\mathbf{p}, s) := y(r^{-1}(\mathbf{p}, s)) = y(r^{-1}\mathbf{p}, r^{-1}s), \tag{17}$$

where $r^{-1}\mathbf{p}$ spatially rotates the pixels, $r^{-1}s$ cyclically permutes the 4 channels.

The G-space (functions over $p4$) with a natural action of $p4$ on it:

$$[(\mathbf{t}, r).y](\mathbf{p}, s) := y((\mathbf{t}, r)^{-1} \cdot (\mathbf{p}, s)) = y(r^{-1}(\mathbf{p} - \mathbf{t}), r^{-1}s) \tag{18}$$

The group convolution in discrete case is defined as

$$[\psi \star x](g) := \sum_{h \in H} \psi(g^{-1} \cdot h) \, x(h). \tag{19}$$

The group convolution with filter $\psi$ and signal $x$ on $p4$ group is given by:

$$[\psi \star x](\mathbf{t}, r) := \sum_{s \in C_4} \sum_{\mathbf{p} \in \mathbb{Z}^2} \psi((\mathbf{t}, r)^{-1}(\mathbf{p}, s)) \, x(\mathbf{p}, s). \tag{20}$$

Using the fact

$$\psi((\mathbf{t}, r)^{-1}(\mathbf{p}, s)) = \psi(r^{-1}(\mathbf{p} - \mathbf{t}, s)) = [r.\psi](\mathbf{p} - \mathbf{t}, s), \tag{21}$$

the convolution can be equivalently written into

$$[\psi \star x](\mathbf{t}, r) := \sum_{s \in C_4} \left( \sum_{\mathbf{p} \in \mathbb{Z}^2} [r.\psi](\mathbf{p} - \mathbf{t}, s) \, x(\mathbf{p}, s) \right). \tag{22}$$

So $\left( \sum_{\mathbf{p} \in \mathbb{Z}^2} [r.\psi](\mathbf{p} - \mathbf{t}, s) \, x(\mathbf{p}, s) \right)$ can be implemented in usual shift-equivariant convolution CONV2D.

The inner sum $\sum_{\mathbf{p} \in \mathbb{Z}^2}$ is equivalently for the sum in steerable convolution, and the outer sum $\sum_{s \in C_4}$ implement rotation-equivariant convolution that satisfies $H$-steerability kernel constraint. Here, the outer sum is essentially using the *regular* fiber representation of $C_4$.

In other words, group convolution on $p4 = \mathbb{Z}^2 \rtimes C_4$ group is equivalent to steerable convolution on base space $\mathbb{Z}^2$ with the fiber group of $C_4$ with regular representation.

**Stack of feature fields.** Analogous to ordinary CNNs, a feature space in steerable CNNs can consist of multiple feature fields $f_i : \mathbb{Z}^2 \to \mathbb{R}^{c_i}$. The feature fields are stacked $f = \bigoplus_i f_i$ together by concatenating the individual feature fields $f_i$ (along the fiber channel), which transforms under the directly sum $\rho = \bigoplus_i \rho_i$ of individual (fiber) representations. Every layer will be equivariant between input and output field $f_{\text{in}}, f_{\text{out}}$ under induced representations $\pi_{\text{in}}, \pi_{\text{out}}$. For a steerable convolution between more than one-dimensional feature fields, the kernel is matrix-valued [12, 16].

---

[5]Technically, we still need to solve the linear equivariance constraint in Eq. 35 to enable weight-sharing for equivariance, while Weiler and Cesa [16] have implemented it for 2D case.

## H  Symmetric Planning Framework: Additional Details

### H.1  Path planning in neural networks

We provide the detailed construction of doing path planning in neural networks in the Section 4. This further explains the visualization in Figure 2 left.

We use the running example of planning on the 2D grid $\mathbb{Z}^2$. We aim to understand (1) how VIN-style networks embed planning and how its idea generalizes, (2) how is symmetry structure defined in path planning and how could it be injected into such planning networks. Recall that we aim to understand (1) how VIN-style networks embed planning and how its idea generalizes, (2) how is symmetry structure defined in path planning and how could it be injected into such planning networks.

**Path planning as MDPs.**  To answer the above two questions, we first need to understand how a VIN embeds a path planning problem into a convolutional network as some embedded MDP. Intuitively, the embedded MDP in a VIN is different from the original path planning problem, since (planar) convolutions are translation equivariant but there are different obstacles in different regions.

For path planning on the 2D grid $\mathcal{S} = \mathbb{Z}^2$, the objective is to avoid some obstacle region $\mathcal{C}_{\mathrm{obs}} \subset \mathbb{Z}^2$ and navigate to the goal region $\mathcal{C}_{\mathrm{goal}}$ through free space $\mathcal{C} \backslash \mathcal{C}_{\mathrm{obs}}$. An action $a = \Delta s \in \mathcal{A}$ is to move from the current state $s$ to a next *free* state $s' = s + \Delta s$, where for now we limit it to be in four directions: $\mathcal{A} =$. Assuming deterministic transition, the agent moves to $s'$ with probability 1 if $s + \Delta s \in \mathcal{C} \backslash \mathcal{C}_{\mathrm{obs}}$. If it hits an obstacle, it stays at $s$ if $s + \Delta s \in \mathcal{C}_{\mathrm{obs}}$: $P(s + \Delta s \mid s, \Delta s) = 0$ and $P(s \mid s, \Delta s) = 1$. Every move has a constant negative reward $R(s, a) = -1$ to encourage shortest path. We call this *ground* path planning MDP, a 5-tuple $\mathcal{M} = \langle \mathcal{S}, \mathcal{A}, P, R, \gamma \rangle$.

**Constructing embedded MDPs.**  However, such transition function is not translation-invariant, i.e. at different position, the transition probabilities are not related by any symmetry: $P(s'|s, a) \neq P(g.s'|g.s, g.a)$. Instead, we could always construct a "symmetric" MDP that has equivalent optimal value and policy for path planning problems, which is implicitly realized in VINs. The idea is to move the information of obstacles from transition function to reward function: when we hit some action $s + \Delta s \in \mathcal{C}_{\mathrm{obs}}$, we instead allow transition $\bar{P}(s + \Delta s \mid s, \Delta s) = 1$ (with all other $s'$ as 0 probability) while set a "trap" with negative infinity reward $\bar{R}_m(s, \Delta s) = -\infty$. The reward function needs the information from the occupancy map $M$, indicating obstacles $\mathcal{C}_{\mathrm{obs}}$ and free space. For the free region, the reward is still a constant $\bar{R}_M(s, \Delta s) = -1$, indicating the cost of movement.

We call it the *embedded* MDP, with different transition and reward function $\bar{\mathcal{M}} = \langle \mathcal{S}, \mathcal{A}, \bar{P}, \bar{R}_M, \gamma \rangle$, which converts the "complexity" in the transition function $P$ in $\mathcal{M}$ to the reward function $\bar{R}_m$ in $\bar{\mathcal{M}}$. Here, map $M$ shall also be treated as an "input", thus later we will derive how the group acts on the map $g.M$. It has the same optimal policy and value as the ground MDP $\mathcal{M}$, since the optimal policies in both MDPs will avoid obstacles in $\mathcal{M}$ or trap cells in $\bar{\mathcal{M}}$. It could be easily verified by simulating value iteration backward in time from the goal position.

The transition probability $\bar{P}$ of the embedded MDP $\bar{\mathcal{M}}$ is for an "empty" maze and thus translation-invariant. Note that the reward function $\bar{R}$ is not not necessarily invariant. This construction is not limited to 2D grid and generalizes to continuous state space or even higher dimensional space, such as $\mathbb{R}^6$ configuration space for 6-DOF manipulation.

Note, all of this is what we use to conceptually understand how a VIN is possible to learn. The reward cannot be negative infinity, but the network will learn it to be smaller than all desired Q-values.

### H.2  Understanding steerable planning

How do we deal with potential symmetry in path planning? how do we characterize it? We try to understand symmetric planning (steerable planning after integrating symmetry with equivariance) and how it is difference classic planning algorithms, such as A*, for planning under *symmetry*.

---

[5]We avoid the symbol $\pi$ for policy since it is used for induced representation in [14, 16].

**Steerable planning.** Recall that we generalize the idea of VIN by considering it as a planning network that composes of mappings between steerable feature fields.

The critical point is that, convolutions directly operate on local patches of pixels and never directly touch coordinates of pixels. In analogy, this avoids a critical drawback in other *explicit* planning algorithms: in sampling-based planning, a trajectory $(s_1, a_1, s_2, a_2, \ldots)$ is sampled and inevitable represented by states $\Omega = \mathcal{S}$. However, to find another symmetric state $g.s$, we potentially need to compare it against all known states $\mathcal{S}' \subset \mathcal{S}$ with all symmetries $g \in G$. On high level, an implicit planner can avoid such symmetry breaking and is more easily compatible with symmetry by using equivariant constraints.

We can use MDP homomorphism to understand this [3, 32].

**MDP homomorphisms.** An *MDP homomorphism* $h : \mathcal{M} \to \overline{\mathcal{M}}$ is a mapping from one MDP $\mathcal{M} = \langle \mathcal{S}, \mathcal{A}, P, R, \gamma \rangle$ to another $\overline{\mathcal{M}} = \langle \overline{\mathcal{S}}, \overline{\mathcal{A}}, \overline{P}, \overline{R}, \gamma \rangle$ [3, 32]. $h$ consists of a tuple of surjective maps $h = \langle \phi, \{\alpha_s \mid s \in \mathcal{S}\} \rangle$, where $\phi : \mathcal{S} \to \overline{\mathcal{S}}$ is the state mapping and $\alpha_s : \mathcal{A} \to \overline{\mathcal{A}}$ is the *state-dependent* action mapping. The mappings are constructed to satisfy the following conditions:

$$\overline{R}\left(\phi(s), \alpha_s(a)\right) \triangleq R(s,a) ,$$
$$\overline{P}\left(\phi\left(s'\right) \mid \phi(s), \alpha_s(a)\right) \triangleq \sum_{s'' \in \phi^{-1}(\phi(s'))} P\left(s'' \mid s, a\right) , \tag{23}$$

for all $s, s' \in \mathcal{S}$ and for all $a \in \mathcal{A}$.

We call the *reduced* MDP $\overline{\mathcal{M}}$ the *homomorphic image* of $\mathcal{M}$ under $h$. If $h = \langle \phi, \{\alpha_s \mid s \in \mathcal{S}\} \rangle$ has *bijective* maps $\phi$ and $\{\alpha_s\}$, we call $h$ an *MDP isomorphism*. Given MDP homomorphism $h$, $(s, a)$ and $(s', a')$ are said to be $h$-equivariant if $\sigma(s) = \sigma(s')$ and $\alpha_s(a) = \alpha_{s'}(a')$.

**Symmetry-induced MDP homomorphisms.** Given group $G$, an MDP homomorphism $h$ is said to be *group structured* if any state-action pair $(s, a)$ and its transformed counterpart $g.(s, a)$ are mapped to the same abstract state-action pair: $(\phi(s), \alpha_s(a)) = (\phi(g.s), \alpha_{g.s}(g.a))$, for all $s \in \mathcal{S}, a \in \mathcal{A}, g \in G$. For convenience, we denote $g.(s, a)$ as $(g.s, g.a)$, where $g.a$ implicitly[6] depends on state $s$. Applied to the transition and reward functions, the transition function $P$ is $G$-invariant if $P$ satisfies $P(g.s'|g.s, g.a) = P(s'|s, a)$, and reward function $R$ is $G$-invariant if $R(g.s, g.a) = R(s, a)$, for all $s \in \mathcal{S}, a \in \mathcal{A}, g \in G$.

However, this only fits the type of symmetry in [9, 10]. And also, they cannot handle invariance to translation $\mathbb{Z}^2$. In our case, we need to augment the reward function with map $M$ input:

$$R_{g.M}(g.s, g.a) = R_M(s, a), \tag{24}$$

for all $s \in \mathcal{S}, a \in \mathcal{A}, g \in G = p4m$.

This means that, at least for rotations and reflections $D_4$, the MDPs constructed from transformed maps $\{g.M\}$ are MDP *isomorphic* to each other.

# I   Symmetric Planning Framework: Proofs

We show the derivation and proofs for all theoretical results in this section.

We follow the notation in [12] to use $\star$ for (one-argument) convolution and $\cdot$ for (two-argument) multiplication:

$$E^a[V](s) = [P^a \cdot V](s) \equiv \sum_{s'} P^a\left(s' \mid s\right) \cdot V(s') \tag{25}$$

## I.1   Proof: equivariance of scalar-valued expected value operation

We present the Theorem 4.1 here and its formal definition.

---

[6]The group operation acting on action space $\mathcal{A}$ *depends on state*, since $G$ actually acts on the *product space* $\mathcal{S} \times \mathcal{A}$: $(g, (s, a)) \mapsto g.(s, a)$, while we denote it as $(g.s, g.a)$ for consistency with $h = \langle \phi, \{\alpha_s \mid s \in \mathcal{S}\} \rangle$. As a bibliographical note, in van der Pol et al. [32], the group acting on state and action space is denoted as state transformation $L_g : \mathcal{S} \to \mathcal{S}$ and *state-dependent* action transformation $K_g^s : \mathcal{A} \to \mathcal{A}$.

**Theorem I.1.** *If transition is $G$-invariant, the expected value operator $E$ over $\mathbb{Z}^2$ is $G$-equivariant:*

$$[g.E^a[V]]\,(s) = [E^{g.a}[g.V]]\,(s), \quad \text{for all } g = tr \in \mathbb{Z}^2 \rtimes D_4.$$

*Proof.* $E$ is the expected value operator. We also write the transition probability as

Recall the $G$-invariance condition of transition probability, the group element $g$ acts on $s, a, s'$:

$$\bar{P}(s' \mid s, a) = \bar{P}(g.s' \mid g.s, g.a) \equiv \bar{P}\left((tr).s' \mid (tr).s, r.a\right), \quad \forall g = tr \in \mathbb{Z}^2 \rtimes D_4, \forall s, a, s', \tag{26}$$

where we can uniquely decompose any $g \in \mathbb{Z}^2 \rtimes D_4$ as $t \in \mathbb{Z}^2$ and $r \in D_4$ [14]. Note that, since the action is the difference between states $a = \Delta s = s' - s$, the translation part $t$ acts trivially on it, so $g.a = (tr).a = r.a$ for all $r \in D_4$.

We transform the feature field and show its equivariance:

$$[g.E^a[V]](s) \equiv [g.[P^a \cdot V]](s) \tag{27}$$

$$\equiv \sum_{s'} \rho_{\text{triv}}(r) P^a \left(s' \mid (tr)^{-1}.s\right) \cdot V(s') \tag{28}$$

$$= \sum_{s'} \rho_{\text{triv}}(r) P^{r.a} \left((tr).s' \mid s\right) \cdot V(s') \tag{29}$$

$$= \sum_{\tilde{s}'} \rho_{\text{triv}}(r) P^{r.a} \left(\tilde{s}' \mid s\right) \cdot V \left((tr)^{-1}\tilde{s}'\right) \tag{30}$$

$$= \sum_{\tilde{s}'} P^{r.a} \left(\tilde{s}' \mid s\right) \cdot \rho_{\text{triv}}(r) V \left((tr)^{-1}\tilde{s}'\right) \tag{31}$$

$$\equiv [P^{r.a} \cdot [g.V]](s) \tag{32}$$

$$\equiv [E^{r.a}[g.V]](s). \tag{33}$$

We use the trivial representation $\rho_{\text{triv}}(g) = \text{Id}_{1 \times 1} = 1$ to emphasize that (1) the group element $g$ acts on *feature fields* $P^a$ and $V$, and (2) both feature fields $P^a$ and $V$ are scalar-valued and correspond to the one-dimensional trivial representation of $r \in D_4$.

In the third line, we use the $G$-invariance of transition probability.

The fourth line uses substitution $\tilde{s}' \triangleq (tr).s'$, for all $s' \in \mathbb{Z}^2$ and $tr \in \mathbb{Z}^2 \rtimes D_4$. This is an one-to-one mapping and the summation does does not change.

$\square$

## I.2  Proof: *expected value operator* as steerable convolution

In this section, we derive how to cast expected value operator as steerable convolution. The equivariance proof is in the next section.

In Theorem 4.1, we show equivariance of value iteration in 2D path planning, while it is only for the case that feature fields $P^a$ and $V$ are scalar-valued and correspond to one-dimensional trivial representation of $r \in D_4$.

Here, we provide the derivation for Theorem 4.2 show that steerable CNNs [14] can achieve value iteration since we could construct the G-invariant transition probability as a steerable convolutional kernel. This generalizes Theorem 4.1 from scalar-valued kernel (for transition probability) with trivial representation to matrix-valued kernel with any combination of representations, enabling using stack (direct-sum) of feature fields and representations.

We state Theorem 4.2 here for completeness:

**Theorem I.2.** *If transition is $G$-invariant, there exists a (one-argument, isotropic) matrix-valued steerable kernel $P^a(s - s')$ (for every action), such that the expected value operator can be written as a steerable convolution and is $G$-equivariant:*

$$E^a[V] = P^a \star V, \quad [g.[P^a \star V]](s) = [P^{g.a} \star [g.V]](s), \quad \forall s \in \mathbb{Z}^2, \forall g \in \mathbb{Z}^2 \rtimes D_4. \tag{34}$$

**Steerable kernels.** In our earlier definition, $\psi^a$ and $f_{\text{in}}$ are transition probability and value function, which are both real-valued $\psi^a : \mathbb{Z}^2 \to \mathbb{R}, f_{\text{in}} : \mathbb{Z}^2 \to \mathbb{R}$. However, this is a *special case* which corresponds to use one-dimensional *trivial representation* of the fiber group $D_4$. In the general case in steerable CNNs [14, 16], we can choose the feature fields $\psi^a : \mathbb{Z}^2 \to \mathbb{R}^{C_{\text{out}} \times C_{\text{in}}}$ and $f_{\text{in}} : \mathbb{Z}^2 \to \mathbb{R}^{C_{\text{in}}}$ and their fiber representations, which we will introduce the group representations of $D_4$ and how to choose in practice in the next section.

Weiler et al. [54] show that *convolutions* with *steerable kernels* $\psi^a : \mathbb{Z}^2 \to \mathbb{R}^{C_{\text{out}} \times C_{\text{in}}}$ is the most general *equivariant linear map* between steerable feature space, transforming under $\rho_{\text{in}}$ and $\rho_{\text{out}}$. In analogy to the continuous version[7] in [16], the convolution is equivariant *iff* the kernel satisfies a $H$-steerability kernel constraint:

$$\psi^a(hs) = \rho_{\text{out}}(h)\psi^a(s)\rho_{\text{in}}(h^{-1}) \quad h \in H = D_4, s \in \mathbb{Z}^2. \tag{35}$$

**Expected value operation as steerable convolution.** The foremost step is to show that the expected value operation is a form of convolution and is also $G$-equivariant. By definition, if we want to write a (linear) operator as a form of convolution, we need one-argument kernel. Cohen et al. [12] show that every linear equivariant operator is some convolution and provide more details. For our case, this is formally shown as follows.

**Proposition I.3.** *If the transition probability is $G$-invariant, it can be expressed as an (one-argument) kernel $P^a(s'|s) = P^a(s' - s)$ that only depends on the difference $s' - s$.*

*Proof.* The form of our proof is similar to [12], while its direction is different from us. We construct a MDP such that the transition probability kernel is $G$-invariant, while Cohen et al. [12] assume the linear operator $\psi \cdot f$ is linear *equivariant* operator on a homogeneous space, and then derive that the kernel is $G$-invariant and expressible as one-argument kernel. Additionally, our kernel $\psi^a(s, s')$ and $\psi^a(s - s')$ both live on the base space $B = \mathbb{Z}^2$ but not on the group $G = \mathbb{Z}^2 \rtimes D_4$.

We show that the transition probability only depends on the difference $\Delta s = s' - s$, so we can define the two-argument kernel $P^a(s'|s)$ on $\mathcal{S} \times \mathcal{S}$ by an one-argument kernel $P^a(s' - s)$ (for every action $a$) on $\mathcal{S} = \mathbb{Z}^2$, without loss of generality:

$$P^a(s' - s) \equiv P^a(\mathbf{0}, s' - s) \tag{36}$$

$$= P^{g.a}(g.\mathbf{0}, g.(s' - s)) \tag{37}$$

$$= P^{r.a}((rs).\mathbf{0}, (rs).(s' - s)) \tag{38}$$

$$= P^{r.a}(r.s, r.(s' - s + s)) \tag{39}$$

$$= P^{r.a}(r.s, r.s') \tag{40}$$

$$= P^a(s, s'), \tag{41}$$

where the second step uses $G$-invariance with $g = sr$, understood as the composition of a translation $s \in \mathbb{Z}^2$ and a transformation in $r \in D_4$.

$\square$

Additionally, we can also derive that, for the one-argument kernel, if we rotate state difference $r.(s' - s)$, the probability is the same for rotated action $r.a$.

$$P^a(s' - s) = P^{r.a}(r.(s' - s)), \text{for all } r \in D_4, s, s' \in \mathbb{Z}^2 \tag{42}$$

The *expected value operator* with two-argument kernel can be then written as

$$E[V](s) \equiv [P^a \cdot V](s) = \sum_{s'} P^a(s'|s)V(s') = \sum_{s'} P^a(s' - s)V(s') \equiv [P^a \star V](s). \tag{43}$$

Note that we do not differentiate between cross-correlation $(s' - s)$ and convolution $(s - s')$.

---

[7]Weiler and Cesa [16] use letter $G$ to denote the stabilizer subgroup $H \le \mathrm{O}(2)$ of $\mathrm{E}(2)$.

### I.3 Proof: equivariance of *expected future value*

Our derivation follows the existing work on group convolution and steerable convolution networks [15, 14, 16, 12]. However, the goal of providing the proof is not just for completeness, but instead to emphasize the close connection between how we formulate our planning problem and the literature of steerable CNNs, which explains and justifies our formulation.

Additionally, there are several subtle differences worth to mention. (1) Throughout the paper, we do not discuss kernels or fields that live on a group $G$ to make it more approachable. Nevertheless, group convolutions are a special case of steerable convolutions with fiber representation $\rho$ as regular representation. (2) We use $\mathbb{Z}^2$ as running example. Some prior work uses $\mathbb{R}^2$ or $\mathbb{Z}^2$, but they are merely just differ in integral and summation. (3) The definition of convolution and cross-correlation might be defined and used interchangeably in the literature of (equivariant) CNNs.

**Notation.** To keep notation clear and consistent with the literature [14, 12, 16], we denote the transition probability $\bar{P}(s'|s,a) \triangleq \psi^a(s,s') \in \mathbb{R}$ (one kernel for an action) and value function as $V(s') \triangleq f_{\text{in}}(s') \in \mathbb{R}$, and the resulting expected value as $f_{\text{out}}^a(s) = \sum_{s'} \psi^a(s,s') f_{\text{in}}(s')$ (given a specific action $a$).

**Transformation laws: induced representation.** For some group acting on the base space $\mathbb{Z}^2$, the signals $f : \mathbb{Z}^2 \to \mathbb{R}^c$ are transformed like [14]:

$$[\pi(g)f](x) = f(g^{-1}x) \tag{44}$$

Apply a translation $t$ and a transformation $r \in D_4$ to $f$, we get $\pi(tr)f$. The transformation law on the input space $f_{\text{in}}$ is [14, 16]:

$$f(x) \mapsto [\pi(tr)f](x) \triangleq \rho(r) \cdot \left[ f\left((tr)^{-1}x\right) \right] \tag{45}$$

The transformation law of the output space after applying $\pi_{\text{in}}$ on input $f_{\text{in}}$ is given by [14]:

$$[\psi \star f](x) \mapsto [\psi \star [\pi(tr)f]](x) \triangleq \rho(r) \cdot \left[ [\psi \star f]\left((tr)^{-1}x\right) \right]. \tag{46}$$

In our case, the output space is $f_{\text{out}}^a : \mathbb{Z}^2 \to \mathbb{R}^{C_{\text{out}}}$ and the input space is $f_{\text{in}} : \mathbb{Z}^2 \to \mathbb{R}^{C_{\text{in}}}$. Intuitively, if we rotate a vector field (fibers represent arrows) by the induced representation $\pi(tr)$ of $f$, we also need to rotate the direction of arrows by $\rho(r), r \in D_4$.

**Equivariance.** Now we prove the steerable convolution is equivariant:

$$[\psi^a \star [\pi_{\text{in}}(g)f_{\text{in}}]](s) = [\pi_{\text{out}}(g)f_{\text{out}}^a](s) \quad \forall s \in \mathcal{S}, \forall g \in G. \tag{47}$$

The induced representation of input field $f_{\text{in}}$ is induced by the fiber representation $\rho_{\text{in}}$, expressed by $\pi_{\text{in}} \triangleq \text{ind}_H^G \rho_{\text{in}} = \text{ind}_{D_4}^{\mathbb{Z}^2 \rtimes D_4} \rho_{\text{in}}$, where $\rho_{\text{in}}$ is the fiber representation of group $H = D_4$. The induced representation of output field $\pi_{\text{out}}$ is analogously from $\rho_{\text{out}}$.

Weiler and Cesa [16] proved equivariance of steerable convolutions for $\mathbb{R}^2$ case, while we include the proof under our setup for completeness. The definition in [16] uses a form of *cross-correlation* and we use *convolution*, while it is usually referred to interchangeably in the literature and is equivalent. Cohen and Welling [14], Weiler et al. [54], Weiler and Cesa [16], Cohen et al. [12], Cohen [56] provide more details and we refer the readers to them for more comprehensive account.

The convolution on discrete grids $\mathbb{Z}^2$ with input field $f_{\text{in}}$ transformed by the induced representation $\pi_{\text{in}}$ gives:

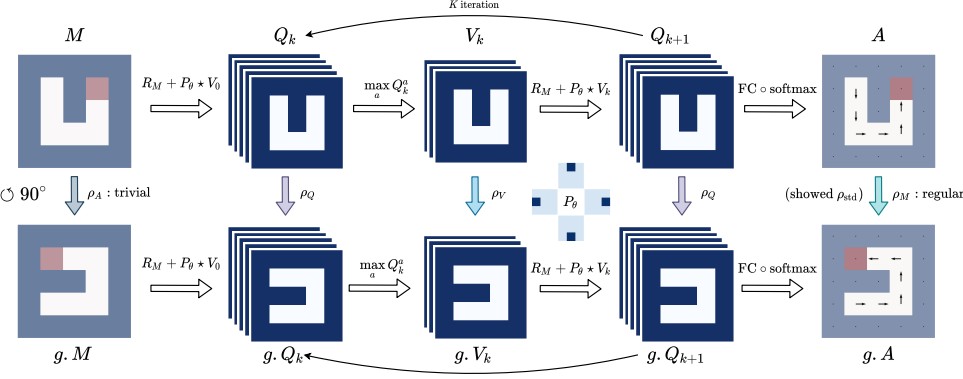

Figure 19: *We attach a copy of the commutative diagram of SymVIN to show the equivariance of steerable value iteration.* Commutative diagram for the full pipeline of SymVIN on steerable feature fields over $\mathbb{Z}^2$ (every grid). If rotating the input map $M$ by $\pi_M(g)$ of any $g$, the output action $A = \texttt{SymVIN}(M)$ is guaranteed to be transformed by $\pi_A(g)$, i.e. the entire steerable SymVIN is equivariant under induced representations $\pi_M$ and $\pi_A$: $\texttt{SymVIN}(\pi_M(g)M) = \pi_A(g)\texttt{SymVIN}(M)$. We use stacked feature fields to emphasize that SymVIN supports direct-sum of representations beyond scalar-valued.

$$
\begin{aligned}
[\psi^a \star [\pi_{\text{in}}(rt)f_{\text{in}}]](s) &= \sum_{s' \in \mathbb{Z}^2} \psi^a(s - s')[\pi_{\text{in}}(rt)f_{\text{in}}](s') \\
&= \sum_{s' \in \mathbb{Z}^2} \psi^a(s - s')\rho_{\text{in}}(r)f_{\text{in}}(r^{-1}(s' - t)) \\
&= \sum_{s' \in \mathbb{Z}^2} \rho_{\text{out}}(r)\psi^a(r^{-1}(s - s'))\rho_{\text{in}}(r)^{-1}\rho_{\text{in}}(r)f_{\text{in}}(r^{-1}(s' - t)) \\
&= \rho_{\text{out}}(r) \sum_{s' \in \mathbb{Z}^2} \psi^a(r^{-1}(s - s'))f_{\text{in}}(r^{-1}(s' - t)) \\
&= \rho_{\text{out}}(r) \sum_{\tilde{s} \in \mathbb{Z}^2} \psi^a(r^{-1}(s - t) - \tilde{s})f_{\text{in}}(\tilde{s}) \\
&= \rho_{\text{out}}(r)f_{\text{out}}(r^{-1}(s - t)) \\
&= [\pi_{\text{out}}(rt)f_{\text{out}}^a](s),
\end{aligned}
\tag{48}
$$

where $s' \in \mathcal{S} = \mathbb{Z}^2$, and thus satisfies the equivariance condition:
$$
[\psi^a \star [\pi_{\text{in}}(rt)f_{\text{in}}]](s) = [\pi_{\text{out}}(rt)f_{\text{out}}^a](s), \forall s \in \mathbb{Z}^2, \forall rt \in \mathbb{Z}^2 \rtimes D_4. \tag{49}
$$

1. Definition of $\star$

2. Transformation law of the induced representation $\pi_{\text{in}}$ [14, 16]

3. Kernel steerability $\psi^a(s) = \rho_{\text{out}}(h)\psi^a(h^{-1}s)\rho_{\text{in}}(h^{-1})$ [16]

4. Move and cancel

5. Substitutes $\tilde{s} = r^{-1}(s' - t)$, $r^{-1}s' = r^{-1}t + \tilde{s}$, so $r^{-1}(s - s') = r^{-1}(s - t) - \tilde{s}$. Since $r \in D_4$ and $s - s' \in \mathbb{Z}^2$, the result is still in $p4m$, it is one-to-one correspondence $p4m \times \mathbb{Z}^2 \to \mathbb{Z}^2$, and the summation does not change. Weiler and Cesa [16] analogously considers the continuous case, where $D_4$ is orthogonal transformations so the Jacobian is always 1.

6. Definition of $\star$

7. Transform law of the induced representation $\pi_{\text{out}}$

## I.4  Proof: equivariance of steerable value iteration

As the third and final step, we would like to show that the full steerable value iteration pipeline is equivariant under $G = \mathbb{Z}^2 \rtimes D_4$. We need to show that every operation in the steerable value iteration is equivariant.

The key is to prove that $\max_a$ is an equivariant non-linearity over feature fields, which follows Section D.2 in [16].

**Step 1:** $V \mapsto Q$. Here, we prove the equivariance of $Q_k^a(s) = \bar{R}_M^a(s) + \gamma \times [\bar{P}_\theta^a \star V_k](s)$. First, let the group acts on both sides:

$$Q_k^a(s) = \bar{R}_M^a(s) + \gamma \times [\bar{P}_\theta^a \star V_k](s) \tag{50}$$

$$\iff [\pi_{\text{out}}(g)Q_k^a](s) = [\pi_{\text{out}}(g)\bar{R}_M^a](s) + \gamma \times [\pi_{\text{out}}(g)[\bar{P}_\theta^a \star V_k]](s) \tag{51}$$

$$\iff [\pi_{\text{out}}(g)Q_k^a](s) = [\pi_{\text{out}}(g)\bar{R}_M^a](s) + \gamma \times [\bar{P}_\theta^a \star [\pi_{\text{in}}(g)V_k]](s) \tag{52}$$

$$\iff Q_k^{g.a}(g^{-1}s) = \bar{R}_{g.M}^{g.a}(g^{-1}s) + \gamma \times [\bar{P}_\theta^{g.a} \star V_k](g^{-1}s) \tag{53}$$

$$\iff Q_k^{\tilde{a}}(\tilde{s}) = \bar{R}_{\pi_{\text{M}}(g)M}^{\tilde{a}}(\tilde{s}) + \gamma \times [\bar{P}_\theta^{\tilde{a}} \star V_k](\tilde{s}) \tag{54}$$

The the last step we substitute $\tilde{s} = g^{-1}s$ and $\tilde{a} = g.a$.

$M : \mathbb{Z}^2 \to \{0,1\}^2$ is the concatenation of maze occupancy map and goal map, which also lives on $\mathbb{Z}^2$. We use two copies of trivial representations as fiber representation $\rho_{\text{M}}$, and denote the induced representation of the field $M$ as $\pi_{\text{M}}$.

Then, we prove the equivariance: if we transform the occupancy map (and goal map), the value iteration should have both input $V$ and output $Q$ transformed. Since this is an iterative process, the only input to the value iteration is actually the occupancy map $M : \mathbb{Z}^2 \to \{0,1\}^2$.

Before that, we observe that the reward also has $G$-invariance when we have map as input:

$$\bar{R}_M^a(s) = \bar{R}_{g.M}^{g.a}(g.s). \tag{55}$$

Additionally, since the reward $\bar{R}_M^a(s)$ means the reward at given position in map $M$ **after executing action** $a$, when we transform the map, we also need to transform the action: $\bar{R}_{g.M}^{g.a}(s)$.

Since it is iterative process, let the $Q$-map being transformed by $g$:

$$[g.Q_k^a](s) = Q_k^a(g^{-1}s) \tag{56}$$

$$= \bar{R}_M^a(g^{-1}s) + \gamma \times [\bar{P}_\theta^a \star V_k](g^{-1}s) \tag{57}$$

$$= \bar{R}_{g.M}^{g.a}(s) + \gamma \times [\bar{P}_\theta^a \star V_k](g^{-1}s) \tag{58}$$

$$= \bar{R}_{g.M}^{g.a}(s) + \gamma \times [\bar{P}_\theta^{g.a} \star [g.V_k]](s) \tag{59}$$

The second last step uses the $G$-invariance condition $\bar{R}_M^a(s) = \bar{R}_{g.M}^{g.a}(g.s)$. The last step uses the equivariance of steerable convolution.

It should be understood as: (1) transforming map $g.M$ and action $g.a$, is always equal to (2) transforming values $[g.Q_k^a]$ and $[g.V_k]$. This proves the equivariance visually shown in Figure 19.

**Step 2:** $Q \mapsto V$. The second step is to show for $V_{k+1}(s) = \max_a Q_k^a(s)$.

Intuitively, we sum over every channel of each representation. For example, if we have $N$ copies of the regular representation with size $|D_4| = 8$, we transform the tensor $(N \times 8) \times m \times m$ to $(1 \times 8) \times m \times m$ along the $N$ channel. Thus, how we use the $8 \times 8$ regular representation to transform the $N \times 8$ channels still holds for $1 \times 8$, which implies equivariance. The $m \times m$ spatial map channels form the base space $\mathbb{Z}^2$ and are transformed as usual (spatially rotated).

Weiler and Cesa [16] provide detailed illustration and proofs for equivariance of different types of non-linearities.

**Step 3: multiple iterations.** Since each layer is equivariant (under induced representations), Cohen and Welling [15], Kondor and Trivedi [13], Cohen et al. [12] show that stacking multiple equivariant layers is also equivariant. Thus, we know iteratively applying step 1 and 2 (*equivariant steerable Bellman operator*) is also *equivariant* (*steerable value iteration*).

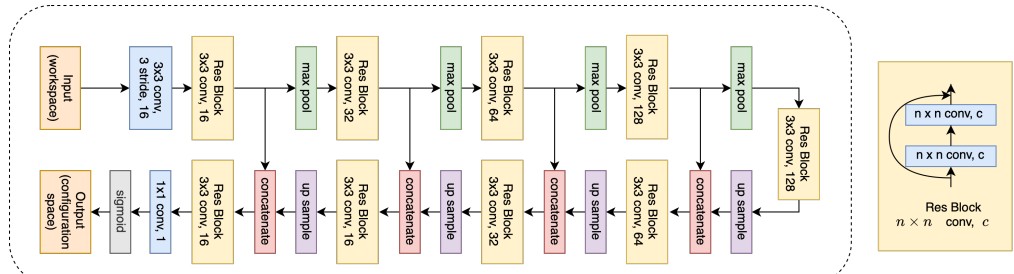

Figure 20: The U-net architecture we used as manipulation mapper.

## J  Practice and Implementation Details

### J.1  Note

We provide additional practical and implementation details, and leave results in the next section.

### J.2  Building Mapper Networks

**For visual navigation.**  For navigation, we follow the setting in GPPN [18]. The input is $m \times m$ panoramic egocentric RGB images in 4 directions of resolution $32 \times 32 \times 3$, which forms a tensor of $m \times m \times 4 \times 32 \times 32 \times 3$. A mapper network converts every image into a 256-dimensional embedding and results in a tensor in shape $m \times m \times 4 \times 256$ and then predicts map layout $m \times m \times 1$.

For the first image encoding part, we use a CNN with first layer of 32 filters of size $8 \times 8$ and stride of $4 \times 4$, and second layer with 64 filters of size $4 \times 4$ and stride of $2 \times 2$, with a final linear layer of size 256.

The second obstacle prediction part, the first layer has 64 filters and the second layer has 1 filter, all with filter size $3 \times 3$ and stride $1 \times 1$.

**For workspace manipulation.**  For **workspace manipulation**, we use U-net [58] with residual-connection [59] as a mapper, see Figure.20. The input is $96 \times 96$ top-down occupancy grid of the workspace with obstacles, and the target is to output $18 \times 18$ configuration space as the maps for planning.

During training, we pre-train the mapper and the planner separately for 15 epochs. Where the mapper takes manipulator workspace and outputs configuration space. The mapper is trained to minimize the binary cross entropy between output and ground truth configurations space. The planner is trained in the same way as described in Section 6.1. After pre-training, we switch the input to the planner from ground truth configuration space to the one from the mapper. During testing, we follow the pipeline in [37] that the mapper-planner only have access to the manipulator workspace.

### J.3  SymGPPN

ConvGPPN [Redacted for anonymous review] is inspired by VIN and GPPN. To avoid the training issues in VIN, GPPN proposes to use LSTM to alleviate them. In particular, it does not use max pooling in the VIN. Instead, it uses a CNN and LSTM to mimic the value iteration process. ConvGPPN, on the other hand, integrates CNN into LSTM, resulting in a single component convLSTM for value iteration. We found that ConvGPPN performs better than GPPN in most cases. Based on ConvGPPN, SymGPPN replaces each convolutional layer with steerable convolutional layer.

### J.4  Understand group conv and "augmented state"

We derive the relationship between group convolution and steerable convolution in Section G.3.

The augmented state $\mathbb{Z}^2 \rtimes D_4 \to \mathbb{R}$ can be similarly treated on the group $p4m = \mathbb{Z}^2 \rtimes D_4$. It is equivalent to using regular representation on the base space $\mathbb{Z}^2$ as $\mathbb{Z}^2 \to \mathbb{R}^8$.

## J.5 Implementation of max operation

Here, we consider how to implement the $\max$ operation in $V_{k+1}(s) = \max_a Q_k^a(s)$. The $\max$ is taken over every state, so the computation mainly depends on our choice of fiber representation.

For example, if we use *trivial representations* for both input and output, the input would be $Q_k : \mathbb{Z}^2 \to \mathbb{R}^{1*C_A}$ and the output is state-value $V_k : \mathbb{Z}^2 \to \mathbb{R}$. This recovers the default value iteration since we take $\max$ over $\mathbb{R}^{C_A}$ vector.

In steerable CNNs, we can use stack of fiber representations. We can choose from regular-regular, trivial-trivial, and regular-trivial (trivial-regular is not considered).

We already covered *trivial* representations for both input and output, they would be $Q_k : \mathbb{Z}^2 \to \mathbb{R}^{C_Q*C_A}$ and $V_k : \mathbb{Z}^2 \to \mathbb{R}^{C_V}$ with $C_Q = C_V = 1$, since every channel would need a trivial representation.

If we use *regular* representation for $Q$ and *trivial* for $V$, they are $Q_k : \mathbb{Z}^2 \to \mathbb{R}^{C_Q*C_A}$ and $V_k : \mathbb{Z}^2 \to \mathbb{R}^{C_V}$ with $C_Q = |D_4| = 8$ and $C_V = 1$. It degenerates that we just take $\max$ over all $C_Q * C_A$ channels.

For both using regular representations, we need to make sure they use the same fiber group (such as $D_4$ or $C_4$), so $C_Q = C_V$. If using $D_4$, we have $Q_k : \mathbb{Z}^2 \to \mathbb{R}^{8*C_A}$ and $V_k : \mathbb{Z}^2 \to \mathbb{R}^8$, and we take max over every $C_A$ channels (for every location) and have 8 channels left, which are used as $\mathbb{Z}^2 \to \mathbb{R}^8$.

Empirically, we found using regular representations for both works the best overall.