# OpenReview forum: "Integrating Symmetry into Differentiable Planning"
_NeurIPS.cc/2022/Conference — NeurIPS 2022 Submitted_

### Official Review · Reviewer_qXsg · 2022-07-08

**Rating:** 6
**Confidence:** 2
**Soundness:** 3 good
**Presentation:** 1 poor
**Contribution:** 2 fair

**Summary:**

The paper proposes using steerable convolutions in the value iteration networks framework to incorporate equivariance under rotations and reflections.

**Questions:**

### Questions

* Does the method extend to rotations that are not $90^\circ$?

* Do you think your method will bring improvements over data augmentation techniques that could extend the data set with rotated/reflected versions of the normal environments?


**Limitations:**

These points are addressed well.

**Strengths And Weaknesses:**

### Strengths

* The proposed approach brings quite a bit of improvement over a regular VIN.
* Equivariance under rotation and reflection are valuable inductive biases.

### Weaknesses

* I believe the paper has an accessibility issue. The main audience for the paper will be researchers interested in VINs. I would expect these to generally not be too familiar with steerible CNNs and the theory that goes into them. Yet, the paper is quite heavy with mathematical notation and terminology from this area. I understand that the authors need to use the math that goes into steerible CNNs to be able to explain their ideas. Though at the moment specificity is getting in the way of clarity. Sections 4 and 5 currently contain too much jargon that is spread throughout the text in a way that makes it genuinely hard to follow the narrative through-line. I would like to give some passages as examples, just to make it clear, what exactly I mean: lines 104-115, 123-128, 171-175, 194-198, 243-249. I think it is in the best interest of the authors to rethink the presentation of sections 4 and 5, and present their ideas in a way that requires a minimum of knowledge about steerible CNNs. Otherwise, you are limiting your reach by creating a hurdle of mathematical prerequisites. Of course, the reader needs to eventually familiarize themselves with the math to truly understand the paper, but the first time reader shouldn't be completely lost either. Again, most people who read your paper won't be familiar with this math, and currently I find it unlikely they will have an easy time.
* The gains in performance over ConvGPPN are somewhat marginal. This would be less of an issue if it weren't for my previous point. If I was a researcher or practitioner interested in using VINs, I believe Table 1 might convince me to use ConvGPPN and accept a slight drop in performance in exchange for the relative algorithmic and theoretical simplicity.

---

> ### Author Response · Authors · 2022-08-02
> **Response to Reviewer qXsg**
>
> We appreciate the reviewer for the time and effort spent on reviewing our work.
> We address the concerns by individual responses, a new section in the appendix on explaining with PyTorch-style pseudocode step-by-step, as well as a new experiment section on generalization to larger maps to demonstrate the significant gap between VIN vs SymVIN and ConvGPPN vs SymGPPN. We uploaded them to the **supplementary material**.
>
> We hope the new pseudocode section can help the reviewer understand from another perspective with minimal prerequisite of equivariant steerable CNNs. We are also open to provide a more intuitive section of the technical section in the next few days if helpful.
>
> **Concern — writing of the technical section. It is hard to understand some concepts, notations and jargons.**
> - Thank you for the feedback on the paper writing. We generally agree that the technical part is not easily accessible and realize this concern is shared with another reviewer.
> - We authors prefer different versions of the technical content (Sec 4+5), and provided a concise version in the main text and a more detailed version in the supplementary material. We wished to provide a more intuitive version for broader audience, while it is hard to do all in the main paper.
> - As a step to solve this, we write a section on explaining the SymVIN method with PyTorch-style pseudocode, since it directly corresponds to what we propose in Section 4 and 5. We try to relate (1) existing concepts with VIN, (2) what we propose in Section 4 and 5 for SymVIN, and (3) actual PyTorch implementation of VIN and SymVIN aligned line-by-line based on semantic correspondence.
> - Thanks to equivariant network community and e2cnn package, the actual implementation of SymVIN is painless and has close relationship with their non-equivariant counterpart. We show two snippets of SymVIN and compare with VIN: the definition of a steerable convolution layer in ~10 lines, and the symmetric value iteration procedure in ~15 lines.
> - We hope this new section can help make terminology more concrete in Section 4 and 5 and demonstrate what actual implementation looks like. We are happy to make the paper more accessible in the future and consider to swap some content in this section with the main text based on further feedback.
> - We will consider to have another short section on intuitively explaining our Symmetric Planning framework and practical considerations in the next few days.
>
> **Concern — ConvGPPN seems good enough. Tasks do seem challenging enough; unknown if algorithms are scalable to them.**
> - **To address this concern, we did new experiment on generalization to larger maps, but we would like to emphasize a few points before going into that.**
>     - We have shown experiments on larger maps in the Section D in appendix (additional result section, moved above, originally at the end). The learning curves of training and validation success rate of SymGPPN and ConvGPPN showed gap between them.
>     - We have done four tasks, all from prior work (VIN, GPPN, SPT and other work along this line [35-39]): (1) 2D path planning (used in VIN, GPPN, SPT, etc), (2) 2DoF C-space manipulation (used in SPT [37]), (3) visual navigation (used in GPPN, SPT [37], etc), (4) workspace manipulation (used in SPT [37]).
>         - For the latter two tasks, since differentiable planning is able to jointly train the transition model with perception module, there is no need for known kinematics/dynamics. This would be intractable for path planning algorithms such as RRT or A*.
>     - We want to highlight that the main algorithm we are studying is SymVIN (vs. VIN), as we use most Section 4 and 5 to explain it. In all experiments, SymVIN clearly outperforms VIN by a large gap. GPPN only empirically does computation of value iteration without theoretical justification, and we developed ConvGPPN and SymGPPN only for completeness. Even though SymGPPN empirically performed the best, it is unclear if the performance gain is due to symmetry in value iteration at all.
>     - Additionally, as we will address for the next question, we already experimented on 50x50, which is larger than VIN and GPPN on 28x28 and match SPT (known for scalability using Transformers) also on 50x50.
> - **To better demonstrate the empirical difference, we conduct new experiment on generalization to larger maps. We hope this can alleviate some concern on (1) scalability and (2) performance gap between SymGPPN and ConvGPPN.**
>     - We experiment all methods on map size 15x15 through 99x99, averaging over 3 seeds (3 model checkpoints, **all trained on 15x15 with K=30**) for each method and 1000 maps for each size. Between 15x15 and 49x49 we use all odd-size maps, and between 51x51 and 99x99 we use interval of 4 (51x51 → 55x55 …).
>     - We keep number of iterations to be K=30 and kernel size F=3 for all methods.
>     - The figure has been added to the new Section B (Figure 8) in the updated appendix in the supplementary material.

---

> > ### Comment · Reviewer_qXsg · 2022-08-04
> > **Thank you for your response**
> >
> > Thank you for your response and the additional supplementary material.
> >
> > **Accessibility of the writing**
> >
> > The pytorch-code shows that your method is easy to implement within the current pytorch ecosystem, which is a definite plus. However, it does not solve the issue that Sections 4 and 5 are hard to follow for a reader unexperienced in steerable convolutions due to how high-level it is.
> >
> > **Comparison to ConvGPPN**
> >
> > Thank you for adding more experiments which show that SymGPPN generalises better than ConvGPPN to larger environments.
> >
> > Currently, my main concern is the presentation of the method, which is not solved by the new pytorch section. If you can make improvements here, I would consider raising my score.

---

> > > ### Author Response · Authors · 2022-08-04
> > > **Thank you for your quick feedback**
> > >
> > > We appreciate your comments on the presentation of Sections 4 and 5. We are currently writing a more intuitive/approachable version of those sections, and plan to add that to the supplementary material in the next few days. We will post another comment when that is done and would highly appreciate your further feedback on that.
> > >
> > > Our current plan is include this simplified version at the beginning of the supplementary material, with a clear note in the main text at the beginning of Section 3 or 4 along the lines of:
> > > "We recommend that readers who are unfamiliar with group equivariance first read the alternative intuitive exposition included in Appendix B, and refer to Sections 4 and 5 on a second read for technical details."
> > >
> > > We have debated whether to replace Sections 4 and 5 in the main text with this simplified version, but we also see that some readers (such as reviewer duzU) find the paper "well-written" and "largely self-contained", so we are also wary of making changes that will compromise technical accuracy and precision. We would appreciate your thoughts on this approach.

---

> > > ### Author Response · Authors · 2022-08-09
> > > **Paper update**
> > >
> > > We uploaded a new revision of the paper. It includes a new intuitive version of the technical sections (method + framework) in appendix Section D, which is written from scratch and contains minimal terminology for equivariant networks / steerable CNNs. We hope this addresses your concern on the writing side.
> > >
> > > We would appreciate any further feedback, and will continue on revising it.

---

> ### Author Response · Authors · 2022-08-02
> **Response part 2 (to technical questions)**
>
> **Question — extension to other rotations?**
> - Yes.
> - Although we have showed that the 2D grid has $D_4$ symmetry, we have tried other symmetry groups in the intermediate layers for the partially equivariant model (since input and output layer must have $D_4$ symmetry, while other layers can be customized). The results are in Figure 13 in L1056 in Appendix Section G.5.
> - It is also possible to extend to the continuous case (on $\mathbb{R}^2$), which is symmetric under rotation group $SO(2)$.
>
> **Question — expecting improvements over data augmentation?**
> - Yes.
> - For our case, 2D maps have no canonical orientation in data generation. Thus, even if we apply data augmentation by random rotations/reflection, a rotated map is still in the distribution of the training data, and the only difference is that the model may see more maps in the same amount of gradient steps.
> - However, if we augment the dataset with all rotations/reflections, this effectively increase the training data size by 8x since non-equivariant models won’t relate them, but this does not contribute to our central goal: computational efficiency. Instead, equivariant methods, such as SymVIN, allow to implicitly plan in a smaller MDP.
> - Furthermore, even if we train on all rotations/reflections of a map, it is not guaranteed to have 0 equivariance error, while injecting equivariance to translation/rotation/reflection to a model can assure this.
> - There are also papers that compare data augmentation methods with equivariance methods. Equivariance methods are always better. Reference:
>     - Wang et al. Data Augmentation vs. Equivariant Networks: A Theory of Generalization on Dynamics Forecasting. arXiv 2022.
>     - Zhu et al. Sample Efficient Grasp Learning Using Equivariant Models. RSS 2022.
> - We also ran new experiments on data augmentation with random rotations/reflections, but didn’t observe significant difference.

---

### Official Review · Reviewer_YbC2 · 2022-07-09

**Rating:** 7
**Confidence:** 1
**Soundness:** 3 good
**Presentation:** 2 fair
**Contribution:** 3 good

**Summary:**

The paper identifies the symmetry in 2D path planning problem and proposes a framework for incoporating the symmetry into an end-to-end differentiable planning framework. This is done mainly by extending the CNN in value iteration network to steerable CNN. The paper demonstrates the advantages of such approaches in two 2D path planning domains.

**Questions:**

1. My understanding is that the symmetry group is manually defined instead of being learned through data. In such case, is there a simpler way of incoporating the symmetry? For example, one way could be applying each transformation in the group, run the planner and returns the plan with the lowest cost.

2. Can the approach generalize to continuous actions?

**Limitations:**

Yes

**Strengths And Weaknesses:**

To say it upfront, with a background in robotic manipulation and reinforcement learning, I do not have background in steerable feature fields and consider the paper outside my area of expertise. I find it difficult to understand the technical details of the paper. As such, I can only provide some high-level feedback to the paper and hope the AC and other reviewers can provide more detailed evaluations.

Here are some suggestions on writing:
1. The explaination of steerable CNN in Sec. 3 heavily references [14, 15, 16], making it hard for me to understand the concept from reading this paragraph alone. Some notations and jargons are also not explained. Figure 2 does not help much as multiple concepts are squeezed into one figure.
2. The benefit of differentiable planning may not be well known in the robotics community. The two tasks done in the paper seems almost trivial and can be easily solved using other path planning techniques like RRT or A*. The paper could benefit from a better motivation on why working on differentiable planning.

---

> ### Author Response · Authors · 2022-08-02
> **Author response to Reviewer YbC2**
>
> We appreciate the reviewer for the time and effort spent on reviewing our work.
> We address the concerns by individual responses and also a new section in the appendix on explaining with PyTorch-style implementation step-by-step.
> We hope the new section can help the reviewer understand from another more concrete perspective.
> We are also open to provide a more intuitive section of the technical section in the next few days if useful.
> We uploaded them to the **supplementary material**.
>
> **Concern — writing of the technical section. It is hard to understand some concepts, notations and jargons.**
> - Thank you for the feedback on the paper writing. We generally agree that the technical part is not easily accessible and realize this concern is shared with another reviewer.
> - We authors prefer different versions of the technical content (Sec 4+5), and provided a concise version in the main text and a more detailed version in the supplementary material. We wished to provide a more intuitive version for broader audience, while it is hard to do all in the main paper.
> - As a step to solve this, we write a section on explaining the SymVIN method with PyTorch-style pseudocode, since it directly corresponds to what we propose in Section 4 and 5. We try to relate (1) existing concepts with VIN, (2) what we propose in Section 4 and 5 for SymVIN, and (3) actual PyTorch implementation of VIN and SymVIN aligned line-by-line based on semantic correspondence.
> - Thanks to equivariant network community and e2cnn package, the actual implementation of SymVIN is painless and has close relationship with their non-equivariant counterpart. We show two snippets of SymVIN and compare with VIN: the definition of a steerable convolution layer in ~10 lines, and the symmetric value iteration procedure in ~15 lines.
> - We hope this new section can help make terminology more concrete in Section 4 and 5 and demonstrate what actual implementation looks like. We are happy to make the paper more accessible in the future and consider to swap some content in this section with the main text based on further feedback.
> - We will consider to have another short section on intuitively explaining our Symmetric Planning framework and practical considerations in the next few days.
>
> **Concern — Why differentiable planning is potentially useful for robotics people? (”The benefit of differentiable planning may not be well known in the robotics community”)**
> - We would like to emphasize that we have done four tasks, all from prior work (VIN, GPPN, SPT and other work along this line [35-39]): (1) 2D path planning (used in VIN, GPPN, SPT, etc), (2) 2DoF C-space manipulation (used in SPT [37]), (3) visual navigation (used in GPPN, SPT [37], etc), (4) workspace manipulation (used in SPT [37]).
>     - We will also edit the paper to make our tasks demonstrated more clear.
> - For the latter two tasks, since differentiable planning is able to jointly train the transition model with perception module, there is no need for known kinematics/dynamics. This would be intractable for path planning algorithms such as RRT or A*.
>     - Concretely, for visual navigation, the input is a collection of 4 egocentric RGB images facing 4 directions (north, east, south, west) in every location, while the workspace manipulation has topdown pixel input.
>     - They are all not typical input to RRT or A* and not trivial to handle. However, they can be easily processed by a perception network first (e.g. a mapper module). Differentiable planning is a known way to be compatible with that, since the planning module and the perception module can be trained together, as shown in our paper and SPT [37].
> - As Reviewer duzU points out, differentiable planning is of interest to the ML community (including NeurIPS), where the above strength (end-to-end differentiability) is one potential reason.

---

> ### Author Response · Authors · 2022-08-02
> **Response part 2 (technical questions)**
>
> **Question — simpler way of incorporating symmetry?**
> - Yes, we have proved what symmetry exists in the (2D) path planning problem and what operation that is (steerable convolution), so we can manually choose the equivariant network to use (steerable CNNs).
> - There are three ways to consider symmetry here: (1) equivariant networks, (2) data augmentation, and (3) “canonicalization” / aggregation over the symmetry group.
> - The method proposed by the reviewer falls into the third category. There are some drawbacks for canonicalization-based methods.
>     - It is hard to consider local symmetry, since for every iteration it needs to apply every transformation in the group, so the total cost is exponential in the planning horizon.
>     - In other words, equivariant methods implicitly plan in a more abstract space/MDP that already quotients out symmetry.
>     - Thus, it increases cost by the scale of the group size (8x for $D_4$), instead of saving computation by planning in a smaller MDP.
>     - It also cannot generalize to continuous case.
> - Additionally, this can be mainly used at inference time, and at training time, applying transformations is equivalent to data augmentation.
> - For data augmentation method, it (1) cannot guarantee 0 equivariance error, and (2) since (2D) maps don’t have canonical orientation, augmentation with rotations/reflections is just effectively increase dataset size.
> - Therefore, equivariance is the best way to consider symmetry in our case.
>
> **Question — generalization to continuous actions?**
> - Yes.
> - One way is to use stochastic transition. We can define that an agent has transition probability proportional to the angle between the continuous action and one of four directions.
> - if we really consider continuous 2D actions, it is possible to work on 2D plane $\mathbb{R}^2$, which has isometries $E(2) \simeq \mathbb{R}^2 \rtimes SO(2)$. There is existing work in $SO(2)$-equivariant networks. For this case, the major issue is to choose the best action in value iteration, which needs to optimize over continuous functions.
> - References
>     - Dian et al., $SO ( 2 )$-Equivariant Reinforcement Learning, ICLR 2022.
>     - Walters et al., Trajectory Prediction using Equivariant Continuous Convolution, ICLR 2021.

---

> ### Author Response · Authors · 2022-08-09
> **Second revision of paper (appendix)**
>
> As a kind reminder, additional to the first revision on adding pseudocode and a new experiment, we just add a new intuitive version of the technical sections (method + framework) in appendix Section D, which is written from scratch and contains minimal terminology for equivariant networks / steerable CNNs. We hope this addresses your concern on the writing side.
>
> We also add a new figure for the generalization experiment (see Figure 8 (right) in **the appendix of the latest supplementary material**), which shows even larger gap between ConvGPPN and SymGPPN.
>
> We would appreciate for any feedback or comments on our response, new sections, and added results.

---

### Official Review · Reviewer_duzU · 2022-07-12

**Rating:** 4
**Confidence:** 3
**Soundness:** 2 fair
**Presentation:** 4 excellent
**Contribution:** 2 fair

**Summary:**

This paper presents an approach to leverage problem-domain symmetry for planning problems defined over small 2D lattices. The key idea is to extend value iteration networks by using steerable convolutions to exploit symmetric structure (e.g., translational/rotational equivariance). Evaluation is carried out on three problem domains, and the presented approach performs better than VIN (value iteration networks) and GPPN (gated path planning networks).

**Questions:**

I would like to see **W1** and **W2** discussed

**Limitations:**

please see **W2** (while authors explicitly list a limitation as an avenue for future work, I believe that limits the scope of the current submission -- this is factored into my eventual score)

**Strengths And Weaknesses:**

Strengths
=========

**S1** The paper is very well-written. Differentiable path planning is an area of great interest to the Neurips(/ICLR/ICML) community, as evidenced by several similar papers in the past (e.g., VIN, GPPN, [35-38]).

**S2** The paper is largely self-contained. As a reader who was not an expert in symmetries and VIN, I appreciated the pointers to relevant readings. This section helped establish context for the technical contributions sections.

**S3** At a technical level, the paper looks well-executed. The core hypothesis is sound -- the 2D gridworld domain (with the considered transition) exhibits symmetries that can be leveraged by learning-based planners to both perform well on the planning problem and also to generalize better to novel problem instances. Building on VINs, which reformulate value iteration as a series of convolutional operators, the proposed approach additionally leverages group convolutions (specifically steerable CNNs) to induce symmetry. The formulation is sound, and achieves better performance compared to variants that do not explicitly assume symmetric structure in the problem domains.

---

Weaknesses
==========

**W1** *Experiments*: Currently, analysis is only carried out on three 2D problem domains (if counting the C-space and workspace manipulation environments as distinct). In table 1, ConvGPPN seems to already achieve stellar performance on all tasks (89.88 success rate % on the workspace manipulation env, >97% on all other envs). The gains due to SymGPPN, while consistent, do not appear to be significant. This might well be because of the inherent task complexity (ConvGPPNs essentially seem to 'solve' the task); to better investigate the benefits of SymGPPN either larger problem instances or more complex problem domains are necessary.

**W2** *Inherent limitations*: One potential reason for choosing low-dimensional (2D) problem domains and further, small problem sizes (largest problem involves 50 x 50 grid) is the (apparent) poor scalability of the approach. Is this because value iteration scales poorly with problem size, and that it is a core component of the approach? Would(n't) this also impact scalability to more complex problem domains (e.g., 3D environments involving larger action spaces, for example)? (While I appreciate the fact that the manuscript lists this as an avenue for future work, I also believe this shortcoming greatly limits the problems this technique can be applied to, thereby affecting the perceived impact of this work).

---

In summary, I think this work tackles an exciting direction; but I am of the opinion that it needs more experimental analysis and some strategies to mitigate the inherent limitations it brings along.

---

> ### Author Response · Authors · 2022-08-02
> **Response to Reviewer duzU**
>
> We appreciate the reviewer for the time and effort spent on reviewing our work.
> We address the concerns by individual responses on why we choose VIN despite of its known scalability issue and also a new experiment section on generalization to larger maps to demonstrate the significant gap between VIN vs SymVIN and ConvGPPN vs SymGPPN.
>
> **Concern — ConvGPPN seems good enough. Tasks do seem challenging enough; unknown if algorithms are scalable to them.**
> - **To address this concern, we did new experiment on generalization to larger maps, but we would like to emphasize a few points before going into that.**
>     - We have shown experiments on larger maps in the Section D in appendix (additional result section, moved up, originally at the end). The learning curves of training and validation success rate of SymGPPN and ConvGPPN showed gap between them.
>     - We have done four tasks, all from prior work (VIN, GPPN, SPT and other work along this line [35-39]): (1) 2D path planning (used in VIN, GPPN, SPT, etc), (2) 2DoF C-space manipulation (used in SPT [37]), (3) visual navigation (used in GPPN, SPT [37], etc), (4) workspace manipulation (used in SPT [37]).
>         - For the latter two tasks, since differentiable planning is able to jointly train the transition model with perception module, there is no need for known kinematics/dynamics. This would be intractable for path planning algorithms such as RRT or A*.
>     - We want to highlight that the main algorithm we are studying is SymVIN (vs. VIN), as we use most Section 4 and 5 to explain it. In all experiments, SymVIN clearly outperforms VIN by a large gap. GPPN only empirically does computation of value iteration without theoretical justification, and we developed ConvGPPN and SymGPPN only for completeness. Even though SymGPPN empirically performed the best, it is unclear if the performance gain is due to symmetry in value iteration at all.
>     - Additionally, as we will address for the next question, we already experimented on 50x50, which is larger than VIN and GPPN on 28x28 and match SPT (known for scalability using Transformers) also on 50x50.
> - **To better demonstrate the empirical difference, we conduct new experiment on generalization to larger maps. We hope this can alleviate some concern on (1) scalability and (2) performance gap between SymGPPN and ConvGPPN.**
>     - We experiment all methods on map size 15x15 through 99x99, averaging over 3 seeds (3 model checkpoints, **all trained on 15x15 with K=30**) for each method and 1000 maps for each size. Between 15x15 and 49x49 we use all odd-size maps, and between 51x51 and 99x99 we use interval of 4 (51x51 → 55x55 …).
>     - We keep number of iterations to be K=30 and kernel size F=3 for all methods.
>     - The figure has been added to the new Section B (Figure 8) in the updated appendix in the supplementary material.

---

> ### Author Response · Authors · 2022-08-02
> **Response part 2**
>
> **Concern — inherent limitations of VIN-based methods.**
>
> In general, we agree that VIN has its shortcomings, and the scalability is exactly the major one we were considering. However, we would like to provide more background on prior work along this line and the reasons behind our choice on VIN despite the scalability concern.
>
> - **"Is this (using low-dimensional problem & small size) because value iteration scales poorly with problem size, and that it is a core component of the approach?"**
>     - We choose this because we follow the prior work (as pointed out, in VIN, GPPN, SPT, and so on). Specifically, VIN mainly experimented on 15x15, and GPPN mainly used 15x15 and tried 28x28 as scalability experiment. SPT advertised to be much better scalable with Transformers and used up to 50x50.
>     - We think integrating symmetry into differentiable planning is an orthogonal topic with the scalability of differentiable planning algorithm, although symmetry could potentially help on scalability.
>     - Also, the new experiment shows that the model can generalizes to larger maps, which unveils potential of scalability. This has not been done in prior work along this line.
> - **Why we choose differentiable planning, specifically VIN, to incorporate symmetry?**
>     - We implement based on value iteration network (VIN) for some reasons.
>     - (1) The expected value operation in value iteration $\sum_{s'} P(s'|s, a) V(s')$ is linear in value function. Since we also proved that value iteration for (2D) path planning is equivariant, this means the Bellman operator is a linear equivariant operator. According to Cohen et al. (2020) [12], any linear equivariant operator has one-to-one correspondence to a (group equivariant) convolution operator.
>     - (2) Value iteration, or Bellman (optimality) operator $V_{k+1}(s) = \max_a R^a(s) + \gamma \times \left[ {P}^a \star V_k \right] (s)$, only relies on operating on fields (“images”) over $\mathbb{Z}^2$, such as value function, reward function, and transition functions.
>         - This enables to inject symmetry (8 states are symmetric under $D_4$) by enforcing same value (after transformation, $D_4$-equivariance), which avoids to find if a new state is symmetric to any existing state.
>     - For the above reasons, we find VIN is empirically the simplest differentiable planning algorithm that satisfies both desiderata.
>     - Additionally, equivariant network community developed techniques to apply convolution networks on non-Euclidean spaces, such as spheres (e.g. spherical CNNs) or even general manifold (gauge equivariant CNNs). It is possible to extend our framework to those cases, which may enable decision-theoretic planning
>         - For example, it is possible to consider planning under uncertainty on a torus formed by a 2-joint arm, which is our experiment on 2DoF C-space/workspace manipulation.

---

> ### Author Response · Authors · 2022-08-09
> **Second revision of paper (appendix)**
>
> As a kind reminder, additional to the first revision on adding pseudocode and a new experiment, we update a new revision of the paper with new intuitive version of the technical sections (method + framework) in appendix Section D.
>
> We also add a new figure for the generalization experiment (see Figure 8 (right) in **the appendix of the latest supplementary material**), which shows even larger gap between ConvGPPN and SymGPPN. Our SymVIN even surpasses ConvGPPN. This figure may further address your concern on the performance of ConvGPPN.
>
> We would appreciate for any feedback or comments on our response, new sections, and added results.

---

### Author Response · Authors · 2022-08-03
**General Response**

- We thank all reviewers for thoughtful and detailed reviews! We found two common concerns from the reviewers: (1) accessibility of technical sections, and (2) benefits of SymGPPN over ConvGPPN.
- We address other concerns and provide further details in individual responses.
- Beyond response, we updated the **(the appendix in) supplementary material** and summarize the modifications to the paper below.
    - (1) A new section on new results and figure.
        - First, we show more concrete visualization to the visual navigation and workspace manipulation tasks, in order to highlight that our SymPlan framework can make use of differentiability to handle sensory input.
        - Second, to address the second concern, we run new experiment on generalization performance of all methods, including ConvGPPN and SymGPPN.
    - (2) A new section on more concrete explanation of SymVIN.
        - This is used to address the first concern. We explain two key steps (defining steerable convolution layer and symmetric VI) of the actual implementation SymVIN with about ~25 lines of code and compare line-by-line with VIN.
    - We also moved the additional experiment/result section from the end of the appendix to Section D now.

---

> ### Author Response · Authors · 2022-08-09
> **Author update on paper revision: a new section of simplified version and a new experiment figure**
>
> Based on the concerns to writing of the technical sections and the discussion with Reviewer qXsg, we wrote a completely new section on the simplified version of the SymPlan method and framework (**Section D in the appendix of the latest supplementary material**). We hope this version provides more intuition and high-level idea. We'll respond to reviewers individually since this has been requested by two different reviewers.
>
> Additionally, we also add a new figure for the generalization experiment (**Figure 8 (right) in the appendix of the latest supplementary material**), which shows even larger gap between ConvGPPN and SymGPPN. Our SymVIN even surpasses ConvGPPN.

---

### Meta-Review · Area_Chair_3xm4 · 2022-08-27

**Recommendation:** Reject
**Confidence:** Certain

**Metareview:**

The paper addresses path planning with RGB inputs by leveraging the workspace symmetry. To that end, the authors propose a end-to-end differentiable planner  that builds on top of VINs and evaluate the method of several 2D-grid planning tasks.

The reviewers recognized that the method presents a performance improvement compared to VINs-like methods, but have raised the questions around the accessibility, scalability, and overall benefits of the method. During the rebuttal, the authors added new experiments to show the method's efficiency in larger environments, and reorganized the manuscript for the better accessibility.

I have read the final version of the manuscript. Based on the current state of the manuscript and the reviewers feedback, I do not believe that it is ready for the publication. My main concerns are around the positioning of the paper and the accessibility.

Positioning of the work -- the authors present the work as addressing robot path planning. The environments and evaluation tasks, even with the new experiments, are toy for the robotics. The 2D C-Space with image observations and no robot dynamics is not a suitable robotics problem. See PRM-RL [Faust et al., ICRA 2018], RL-RRT [Chiang et al., RA-L 2019], Critical PRMs [Ichter et al., ICRA 2020], optimal control w/ visual navigation [Bansal et al, CORL 2020] for methods that combine motion planning, controls and perception (ego sensors, motion planning, and non-trivial robot dynamics and geometry). Granted, they are not differential planning, but they solve more path planning more realistic and complex settings. (In the rebuttal the authors comment that the differentiable planning is capable of jointly training perception with the transition model is intractable for RRT or A*. However, the transition model is trivial here -- there are no kinodynamic constraints, or complicated geometry.) Perhaps a better framing for the presented work is as incorporating symmetry into latent planning, instead of framing it around the robotics.

It is not clear what problem the paper is seeking to solve. Please add a clear definition of the path planning problem. Are the policies goal-conditioned? Is the generalization over the workspaces, or the initial configurations or both? What is in the training set? Are the connections between the planning points known or not? And are there any other constraints on the transition function (beyond the workspace constraints)?

Accessibility --  Even after the rebuttal, Sections 3 and 4 are not clear. The symmetry is not introduced well for a non-expert. Some questions -- If I understand correctly, the symmetric NNs maps inputs to equivalent states. How is that different from latent spaces? Is the proposed method too specific for CNNs, which are rapidly becoming obsolete, in favor of newer models? How would the method compare to VAE? I suggest that authors take an intuitive example of the symmetry (for example, we expect the planner to learn when it sees the wall in a given direction, that the transition in that direction is not possible. The same will hold for left, right, top or bottom. So we hope that by exploiting symmetry, we can speed up the learning since the agent would need to learn only on a single instance of the equivalence class, and generalize to the others.) Lastly, the paper would be stronger, with a more in-depth analysis of the method. Where and how exactly did the symmetry help?

Overall, the exploitation of workspace symmetry in E2E differential planning has merit. But the framing around the robotics, VINs, and CNNs is too specific, yielding results which significance is not clear. With more generalized framing rooted into the current ML trends this paper can make a strong a valuable contribution.

**Award:**

No

---

### Decision · Program_Chairs · 2022-09-14

Reject